# Reasoning Path Compression: Compressing Generation Trajectories for Efficient LLM Reasoning

Jiwon Song [1]    Dongwon Jo [1]    Yulhwa Kim [2*]    Jae-Joon Kim [1*]

[1] Seoul National University    [2] Sungkyunkwan University

{jiwon.song, dongwonjo, kimjaejoon}@snu.ac.kr

{yulhwakim}@skku.edu

## Abstract

Recent reasoning-focused language models achieve high accuracy by generating lengthy intermediate reasoning paths before producing final answers. While this approach is effective in solving problems that require logical thinking, long reasoning paths significantly increase memory usage and reduce throughput of token generation, limiting the practical deployment of such models. We propose Reasoning Path Compression (RPC), a training-free method that accelerates inference by leveraging the semantic sparsity of reasoning paths. RPC periodically compresses the KV cache by retaining cache entries that receive high importance score, which are computed using a selector window composed of recently generated queries. Experiments show that RPC improves generation throughput of QwQ-32B by up to $1.60\times$ compared to the inference with full KV cache, with an accuracy drop of 1.2% on the AIME 2024 benchmark. Our findings demonstrate that semantic sparsity in reasoning traces can be effectively exploited for compression, offering a practical path toward efficient deployment of reasoning LLMs. Our code is available at https://github.com/jiwonsong-dev/ReasoningPathCompression.

## 1   Introduction

Large language models (LLMs) equipped with reasoning capabilities have expanded the application of LLMs beyond simple natural language processing tasks to complex problem-solving tasks such as Science, Technology, Engineering, and Mathematics (STEM) reasoning and code generation. Early reasoning approaches primarily focused on guiding LLMs through explicit step-by-step logic to facilitate more interpretable and accurate outcomes [1]. Recently, advanced reasoning LLMs, such as OpenAI o1 [2], DeepSeek-R1 [3], adopted the concept of *test-time compute scaling* [4, 5]. This method involves generating longer, iterative reasoning outputs, which significantly enhance accuracy. Such iterative generation allows models to carefully evaluate intermediate reasoning steps, refine outputs through internal reflection, and ultimately handle tasks requiring complex reasoning.

Though reasoning LLMs have been widely adopted due to their ability to handle complex tasks through complicated reasoning processes, reasoning LLMs face challenges in inference efficiency due to their tendency to generate long reasoning sequences. The long token sequences required for detailed reasoning processes substantially increase the KV cache overhead during inference. For example, the reasoning path of OpenAI's o3-mini-high can exceed 50K tokens [6], and Claude 3.7 Sonnet [7] supports reasoning sequences of up to 64K tokens. Such a long token generation imposes critical memory and computational overhead, significantly slowing down inference. Consequently, it is crucial to develop KV cache compression techniques to mitigate these inference efficiency issues and support practical deployment of reasoning LLMs.

---

*Corresponding Author

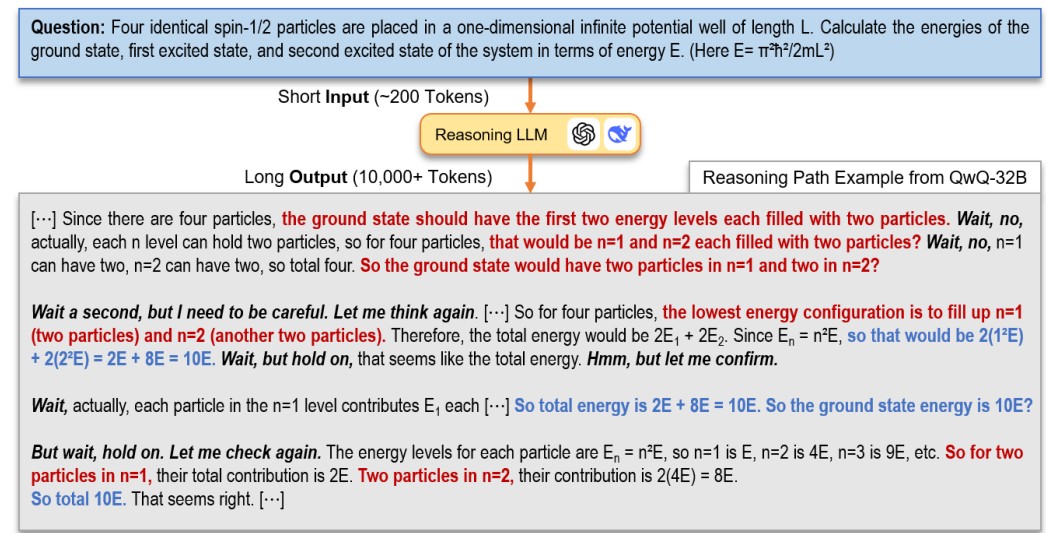

Figure 1: Example of a reasoning path of a reasoning LLM. Redundant reasoning steps (e.g., repeated checks and re-derivations) are visually highlighted, illustrating the semantic sparsity that motivates our compression method. The parts highlighted in the same color are semantically identical.

Although there are several existing works on compressing KV cache for long sequences [8, 9, 10, 11], these works primarily focus on efficient handling of long input prompts. In contrast, the problem of efficiently managing the KV cache for long generated sequences has received limited attention. Unlike input prompts, whose importance can be easily assessed at prefill stage [8], generated tokens pose a challenge because their future relevance is often unpredictable. As a token seemingly insignificant at one point might become crucial later, naively discarding such tokens can substantially degrade model accuracy.

However, as illustrated in Figure 1, we observe that sequences generated during reasoning processes exhibit distinct properties compared to sequences generated in conventional LLM decoding. Specifically, reasoning sequences frequently revisit previous cases or repeat similar logic, so they have low information density relative to their length. We refer to this phenomenon as the *semantic sparsity* of reasoning paths. This sparsity highlights the inefficiency of retaining all KV entries and the possibility to selectively remove KV cache corresponding to less important tokens without disrupting the overall reasoning process.

Motivated by this observation, we propose **Reasoning Path Compression (RPC)**, a method for accelerating inference in reasoning LLMs by compressing the KV cache associated with explicit thinking tokens. RPC compresses KV cache periodically during decoding, significantly reducing overhead compared to previous step-wise compression techniques which compress KV cache at each decoding step. At each compression interval, it estimates token importance based on attention allocation over a recent window and retains only the top-ranked entries according to a fixed compression ratio. This design preserves recent context while discarding low-impact KV entries, mitigating performance degradation. By applying RPC to QwQ-32B [12], we reduce the KV cache size of generated tokens by up to 75%, and improve decoding throughput by up to $1.60\times$, while keeping the pass@1 drop on the AIME 2024 [13] dataset within 1.2% compared to the inference with full KV cache.

## 2 Background

### 2.1 Reasoning LLMs

Reasoning LLMs solve problems by generating explicit intermediate steps, known as reasoning paths, instead of directly producing an answer [2, 3, 12, 14]. This behavior is reinforced by the way such models are trained: reasoning LLMs are typically fine-tuned with reinforcement learning objectives that reward correct answers after multi-step inference, thereby encouraging longer generations. Consequently, the lengths of generated sequences increase as training progresses [15]. Allocating up to 32K tokens for explicit reasoning has yielded steady accuracy gains across complex reasoning

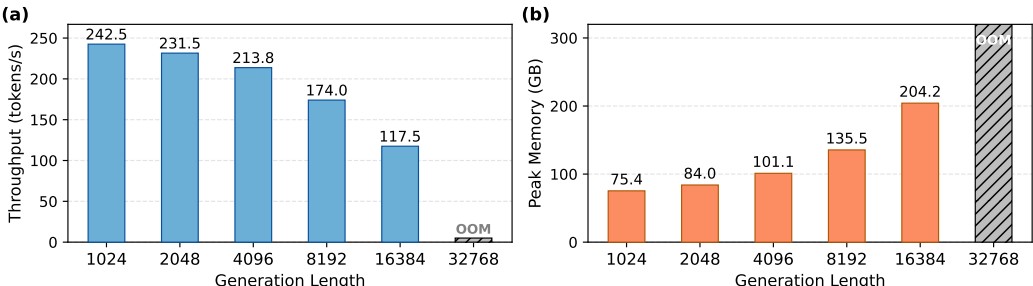

Figure 2: (a) Token generation throughput and (b) peak memory of QwQ-32B at different generation lengths. The results are evaluated on $4\times$ H100 GPUs with batch size 16.

benchmarks without showing signs of plateau, raising expectations of further improvements with even larger thinking budgets [16]. Such extensive token generation significantly enlarges the KV cache, increasing memory usage and reducing inference throughput. As shown in Figure 2, generating sequences of 16K to 32K tokens dramatically reduces throughput while sharply increasing peak memory usage.

To mitigate the overhead of generating long reasoning paths, a large body of work has focused on training-based approaches that encourage reasoning LLMs to produce shorter sequences [17, 18, 19, 20, 21, 22], relatively few attempts have been made to improve reasoning efficiency at inference time [23]. These approaches utilize length-aware training objectives: either encouraging the generation of short sequences or introducing mechanisms to compress tokens into latent representations [24]. However, their effectiveness typically remains limited when applied to complex reasoning benchmarks widely used to evaluate modern reasoning LLMs (e.g., LiveCodeBench [25]). For example, although LightThinker [22] achieves competitive accuracy with shortened reasoning paths on relatively simpler reasoning tasks like MMLU [26] and BBH [27], our experimental results in Section 4.3 indicate a significant performance degradation when evaluated on more complex reasoning benchmarks. This discrepancy arises primarily due to the conflicting training objectives. The reasoning-oriented objectives aim to promote detailed reasoning steps, whereas the length-aware objectives encourage shorter outputs. Thus, effectively training reasoning LLMs to consistently produce shorter reasoning paths remains challenging.

## 2.2 KV Cache Compression

The degradation of throughput and the increase in memory usage observed when processing long sequences with LLMs primarily result from growth in KV cache size. Thus, there are many attempts to directly compress the KV cache, but these works primarily focus on efficient handling of long input prompts. For example, SnapKV [8] and HeadKV [9] are specifically designed to compress KV cache associated with long input contexts. These methods do not address the compression of generated tokens produced in reasoning paths.

Other techniques like H2O [10] and TOVA [11] attempt to extend KV cache compression mechanisms to support basic levels of compression during generation. These methods maintain the KV cache within a predefined budget by evicting tokens whenever the cache reaches this size limit during decoding. However, their designs predominantly target scenarios involving long input sequences and relatively short outputs, they are effective when identifying and evicting less relevant input tokens is critical for efficient output generation. Hence, H2O and TOVA struggle to preserve accuracy when applied to reasoning LLMs (see Section 4.3). Moreover, while setting a fixed KV cache budget is straightforward in input-dominated scenarios, it is challenging to predefine cache budgets for reasoning LLMs, as they inherently produce long output sequences of varying lengths. Overall, there are currently no KV cache compression methods tailored to reasoning LLMs.

## 3 Reasoning Path Compression

### 3.1 Motivation: Semantic Sparsity of Reasoning Paths

Reasoning LLMs do not directly generate the final answer. Instead, they produce reasoning paths, which often contain redundant segments offering little new information, such as repeated logical

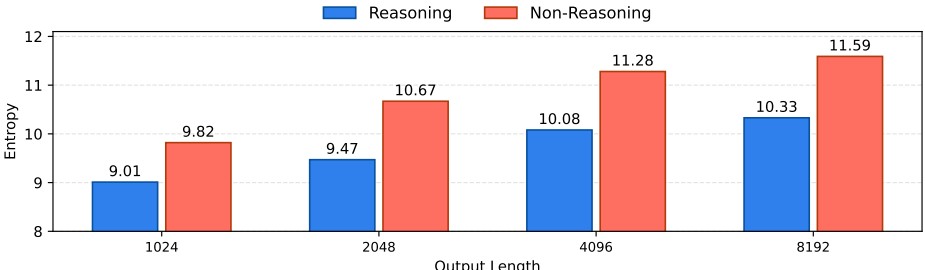

Figure 3: 3-gram Shannon entropy comparison between reasoning LLM and non-reasoning LLM.

steps or re-evaluations of previous generated reasoning. As previously presented in Figure 1, such redundancy is frequently observed in model-generated reasoning paths. Additional examples are provided in Appendix A. We refer to this phenomenon, the presence of extended spans of generated tokens that are semantically redundant, as *semantic sparsity*. To quantify semantic sparsity, we compute the *n-gram Shannon entropy* using base-2 logarithm, defined as:

$$H_n = - \sum_{g \in \mathcal{G}_n} p(g) \log_2 p(g) \tag{1}$$

where $\mathcal{G}_n$ denotes the set of all unique n-grams of length $n$, and $p(g)$ is the empirical probability of each n-gram $g$.

To analyze semantic sparsity of reasoning paths, we compare the redundancy in sequences generated by conventional LLMs and reasoning LLMs. For this comparison, we use 3-gram entropy to measure phrase-level repetition and evaluate two models with identical architecture (LLaMA-3.1-8B-Instruct [28]): DeepSeek-R1-Distill-Llama-8B [3], a reasoning-oriented model, is tested on AIME 2024 [13], and LongWriter-8B [29], tuned for long-form writing, is tested on a subset of HelloBench [30] consisting of prompts that require generating outputs exceeding 8192 tokens.

As shown in Figure 3, DeepSeek-R1-Distill-Llama-8B consistently exhibits lower 3-gram entropy than LongWriter-8B across output lengths from 1024 to 8192 tokens. This indicates higher phrase-level repetition in reasoning paths compared to general long-form writing. These results provide quantitative evidence of semantic sparsity, suggesting that large portions of the reasoning trace can be compressed with minimal impact on overall coherence.

## 3.2 Overview of Reasoning Path Compression

We introduce *Reasoning Path Compression* (RPC), a KV cache compression framework tailored for reasoning LLMs (Figure 4). RPC leverages the semantic sparsity inherent in reasoning paths to efficiently eliminate KV entries. The key insight motivating RPC is that reasoning LLMs generate explicit reasoning steps, and many of these reasoning steps lose relevance as the reasoning process progresses. Exploiting this observation, RPC periodically compresses redundant KV entries during

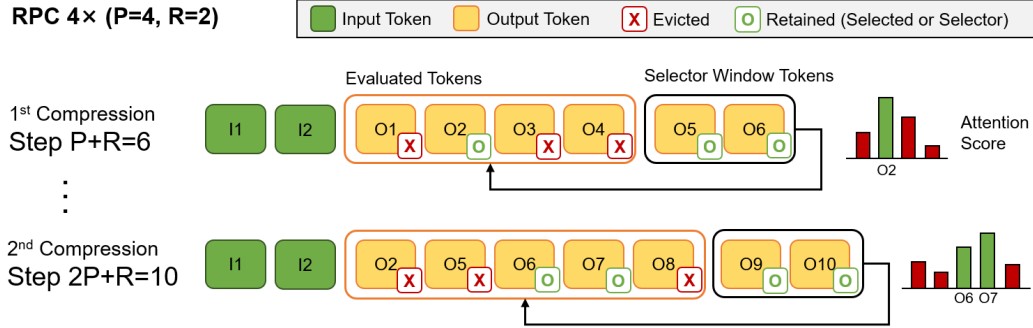

Figure 4: Illustration of RPC with compression interval $P = 4$, selector window $R = 2$, and compression ratio $c = 4$. At each compression step, recent $R$ tokens are used to evaluate the importance of previously generated tokens.

token generation. Moreover, since recently generated tokens inherently rely on the context provided by preceding tokens, these recent tokens serve as essential indicators of contextual importance. Thus, RPC assesses the relevance of previously generated tokens by analyzing how strongly they are attended to by the most recent tokens, referred to as *selector window*. All compression decisions are made dynamically during inference based solely on attention-derived statistics. Hence, RPC does not require any model modification or additional training, and it is straightforward to integrate RPC into existing inference pipelines of reasoning LLMs.

### 3.3  Periodic KV Cache Compression Dynamics of RPC

One of the unique features of RPC compared to other KV cache compression methods is its periodic approach to KV cache compression. The KV cache compression dynamics of RPC is controlled by two critical hyperparameters: the compression interval $P$, which represents how frequently KV cache compression is triggered, and the size of the selector window $R$, which denotes the number of recent tokens used to assess importance.

As illustrated in Figure 4, RPC waits for $P + R$ tokens to be generated to start the first compression cycle. At this point, the importance of the initial set of $P$ tokens is evaluated using the selector window composed of the most recent $R$ tokens. Given a target compression ratio $c$, RPC retains only the top $\frac{P}{c}$ tokens based on their importance scores. Subsequent compression cycles are triggered each time an additional $P$ tokens have been generated.

It is important to note that during each periodic compression cycle, RPC evaluates a combined set comprising both tokens retained from the previous cycle and newly generated tokens, rather than compressing only newly generated ones. By jointly reassessing all these tokens at each cycle, RPC naturally allows outdated tokens to fade out as the reasoning path progresses. As a result, the reasoning context remains properly updated and relevant throughout the inference process, even after multiple cycles of KV cache compression.

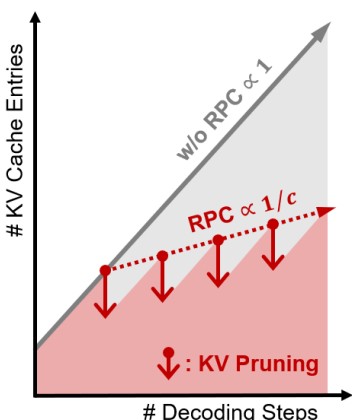

Figure 5: KV cache size with and without RPC.

Specifically, at the second compression cycle, the selector window, now updated to include the latest $R$ tokens, evaluates the importance of the previously retained $\frac{P}{c}$ tokens and the newly generated set of $P$ tokens. Among these $\frac{P}{c} + P$ tokens, RPC retains only the top $\frac{2P}{c}$ tokens with the highest importance scores and discards the rest. Generalizing this procedure, at the $N$-th compression cycle, the total number of tokens evaluated with the selector window is $\frac{(N-1)P}{c} + P$. RPC retains only the top $\frac{NP}{c}$ tokens with the highest importance scores from this set. As selector tokens are always preserved, the total number of KV entries remaining after the $N$-th compression cycle is $\frac{NP}{c} + R$. As shown in Figure 5, this periodic compression effectively regulates the size of KV cache over time.

To fully leverage the advantages of periodic compression, the compression interval $P$ must be carefully selected. A small $P$ value may lead to accuracy degradation after compression, as the semantic context is too limited. On the other hand, a large $P$ provides a broader semantic context for effective compression, while it introduces computational inefficiency and higher peak memory usage by delaying the compression. Given the significance of the compression interval $P$, an ablation study analyzing its impact and recommendations derived from the analysis are discussed in Section 4.5.

### 3.4  Important Token Selection with Selection Window

Another unique feature of RPC is the concept of the selector window used for selecting important tokens. Previous KV cache compression methods employ various strategies for calculating token importance. For example, SnapKV computes attention scores relative to the final tokens in the input prompt, based on the observation that the last segment of the input shows similar attention allocation pattern to the generation stage. H2O averages attention scores across all preceding tokens, and TOVA mimics RNN operations by reusing the attention scores calculated during token generation as gating

**Algorithm 1:** Important token selection algorithm of RPC

---

**Input:** generation step $t$, query of step $t$ $q_t$, KV cache $C_{KV}$, selector query cache $C_{\mathcal{Q}}$
**Output:** updated $C_{KV}$, updated $C_{\mathcal{Q}}$

---

```
// Cache selector queries
```
**if** $(t - R) \geq 0$ **and** $(t - R) \bmod P < R$ **then**
    ⌊ Append $q_t$ to $C_{\mathcal{Q}}$

```
// Compress KV cache every P steps
```
**if** $(t - R) \geq 0$ **and** $(t - R) \bmod P = 0$ **then**
    $s \leftarrow$ *Importance* of tokens in $C_{KV}$ ;         `// Compute importance score`
    $C_{tmp} \leftarrow$ KV cache with top-$\frac{N \cdot P}{c}$ importance scores ; `// Retain important KV cache`
    $C_{KV} \leftarrow C_{tmp} \cup C_{KV}[-R:]$ ;         `// Retain KV cache of selector window`
    $C_{\mathcal{Q}} \leftarrow []$ ;                                 `// Reset selector query cache`

**return** $C_{KV}, C_{\mathcal{Q}}$

---

scores for the KV cache eviction. In contrast, RPC leverages the observation that recently generated tokens in reasoning paths represent logical outputs derived from preceding contexts. Therefore, attention scores relative to these recent tokens can effectively indicate the relevance of previously generated tokens.

The algorithm for selecting important tokens in RPC is presented in Algorithm 1. RPC evaluates token importance using attention scores aggregated across a selector window of the $R$ most recent tokens and all attention heads. Then, to promote coherent token selection and reduce token-level noise, RPC applies local average pooling. Formally, the importance of each past token $t$ at each layer is defined as:

$$Importance(t) = \frac{1}{2w + 1} \frac{1}{R \cdot H} \sum_{i=-w}^{w} \sum_{r=1}^{R} \sum_{h=1}^{H} \text{Attn}_h^\ell(q_r, k_{t+i}) \tag{2}$$

Here, $\text{Attn}_h^\ell(q_r, k_{t+i})$ denotes the attention weight from the $r$-th selector token to token generated at $t + i$-th generation step at head $h$ of layer $\ell$. The pooling window size $w$ controls the smoothing level, encouraging contiguous retention of semantically related tokens. To eliminate redundant computations and efficiently compute these importance scores, RPC caches the query vectors of selector tokens.

The selector window size $R$ determines how many recent tokens RPC uses to assess the importance of previously generated tokens. A smaller $R$ may lead to unstable or noisy importance estimations, as scoring can be dominated by a limited number of tokens. In contrast, larger values of $R$ increase memory overhead by requiring additional caching of query vectors. Thus, choosing an appropriate value for $R$ involves balancing the robustness in token scoring with computational overhead. A detailed ablation study and recommendation for optimal $R$ values are provided in Section 4.5.

## 4 Experiments

### 4.1 Experimental Setup

**Models and Datasets.** We evaluate RPC using two open-source reasoning LLMs with different model sizes: DeepSeek-R1-Distill-Qwen-7B with 7B parameters [3] and QwQ-32B with 32B parameters [12]. All outputs are generated using nucleus sampling with temperature $= 0.6$ and top-$p = 0.95$. For QwQ-32B, we additionally set top-$k = 40$ following the model's recommended decoding configuration. The maximum number of generated tokens is capped at 32768, following the default settings of tested models.

**Datasets.** Our evaluation covers three reasoning-intensive benchmarks: American Invitational Mathematics Examination (AIME) 2024 for mathematical reasoning, LiveCodeBench [25] for coding tasks, and IFEval [31] for instruction following. We sample $k$ completions per instance to compute pass@1, where $k = 8$ for AIME 2024, $k = 4$ for LiveCodeBench, and $k = 1$ for IFEval, respectively.

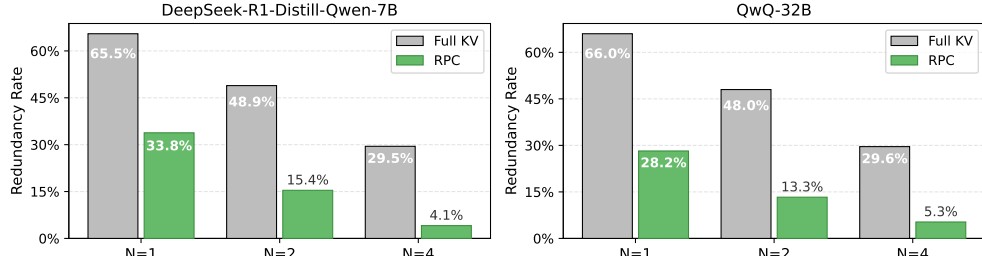

Figure 6: Redundancy rate comparison between full KV and RPC.

Table 1: Accuracy (%) comparison between baselines and RPC.

| Method | DeepSeek-R1-Distill-Qwen-7B | | | QwQ-32B | | |
| --- | --- | --- | --- | --- | --- | --- |
| | AIME 2024 (pass@1) | LiveCodeBench (pass@1) | IFEval (pass@1) | AIME 2024 (pass@1) | LiveCodeBench (pass@1) | IFEval (pass@1) |
| Full KV Cache | 55.5 | 37.6 | 55.1 | 79.5 | 63.4 | 83.9 |
| H2O | 42.5 | 22.5 | 51.8 | 75.0 | 54.2 | 74.3 |
| TOVA | 42.5 | 21.5 | 48.8 | 70.0 | 43.8 | 50.6 |
| LightThinker | 6.7 | 0.7 | 25.1 | - | - | - |
| **RPC** ($P = 4096$) | 52.9 | 35.9 | 56.6 | 78.3 | 62.2 | 82.6 |
| **RPC** ($P = 1024$) | 50.4 | 33.5 | 57.3 | 78.3 | 61.2 | 81.7 |

**Implementation Details.** Our implementation uses FlashAttention-2 [32] as the attention kernel for all decoding layers and is built on top of HuggingFace Transformers v4.45 [33]. Unless otherwise specified, we use the following default RPC hyperparameters: We set the selector window size $R$ to 32 and apply local pooling with window size $w = 3$ for importance smoothing. The compression interval $P$ is set to 1024 or 4096. The target compression ratio is set to $4\times$ by default.

**Baselines.** We compare our proposed RPC with a training-based reasoning path compression method, LightThinker [22], and previous KV cache compression techniques, H2O [10] and TOVA [11]. To ensure a fair comparison with H2O and TOVA, we set their KV cache budgets to match the overall compression ratio ($4\times$) of RPC. For each of the evaluation datasets, we profile the average generation length of the original reasoning LLMs with full KV caches, and allocate 25% of this average length as a fixed KV cache budget for all prompts within that dataset. Meanwhile, LightThinker does not offer direct control over the compression ratio, so we measure its effective compression ratio after inference.

## 4.2 Redundancy Reduction

We quantitatively evaluate the redundancy-reducing effect of RPC using an embedding-based similarity analysis. For each model, we generate outputs on the AIME 2024 dataset using both the full KV baseline and RPC. The generated outputs are segmented into sentences and pairwise cosine similarities are computed between all sentence embeddings within the same output. Two sentences are considered semantically similar if their cosine similarity exceeds 0.75. We define the *redundancy rate* as the proportion of sentences that have more than $N$ semantically similar counterparts within the same output, where $N \in \{1, 2, 4\}$.

As shown in Figure 6, the redundancy rate significantly decreases after applying RPC, across all models. Specifically, the proportion of semantically repetitive sentences (i.e., $N$=1) is reduced by nearly half, and the gap widens for higher redundancy thresholds ($N$=2, 4). This indicates that RPC not only removes verbatim repetitions but also suppresses subtle paraphrased duplications that frequently appear in reasoning trajectories. These results provide strong evidence that RPC effectively leverages semantic sparsity to maintain concise yet coherent reasoning sequences.

Additional details of the embedding-based redundancy analysis and visualized examples of RPC's token selection are provided in Appendix B.

## 4.3 Accuracy Evaluation

Table 1 compares the accuracy between RPC and the baseline methods. LightThinker, a training-based reasoning path compression approach, shows the lowest accuracy across all benchmarks despite

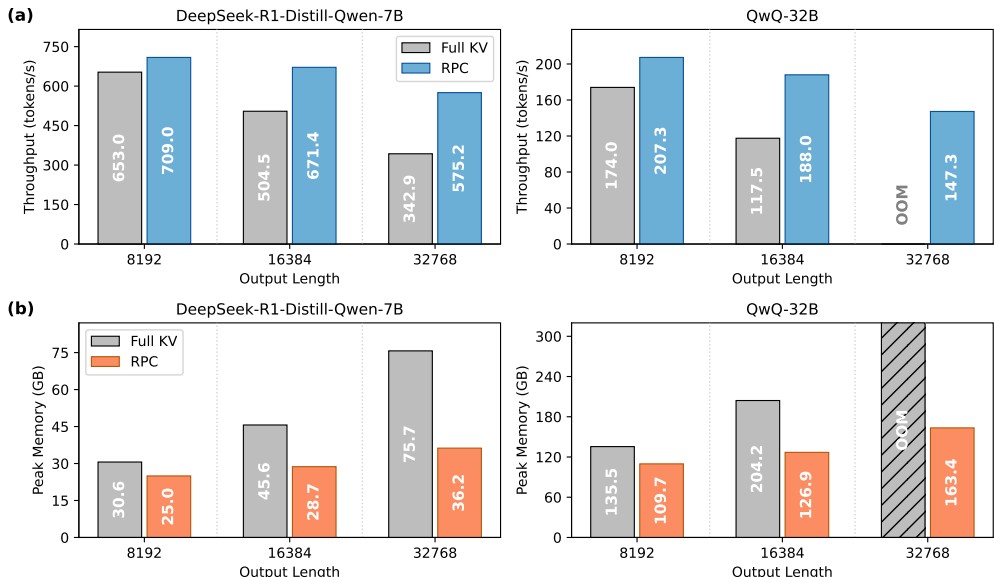

Figure 7: (a) Throughput (tokens/s) and (b) peak memory usage (GB) comparison between RPC ($P = 4096$) and full KV cache inference.

operating with mild compression ratio (1.49×, 1.41×, 1.57× for AIME 2024, LiveCodeBench, IFEval, respectively). This result highlights the limited effectiveness of length-aware training approaches for reasoning LLMs.

Other KV cache compression baselines, such as H2O and TOVA, achieve higher accuracy than LightThinker but still exhibit a significant gap compared to full KV inference. Moreover, their requirement of a predefined KV cache budget limits applicability in real-world scenarios where the total generation length cannot be known in advance.

In contrast, RPC achieves accuracy comparable to full KV inference without any additional training or prior knowledge of the output length. With a compression interval of $P = 4096$, RPC successfully limits the accuracy drop to within 2.6% for DeepSeek-R1-Distill-Qwen-7B and 1.2% for QwQ-32B on AIME 2024. A shorter interval ($P = 1024$) slightly reduces accuracy while providing stronger compression and efficiency. Therefore, careful selection of $P$ is important, and we provide an ablation study analyzing its impact in Section 4.5.

## 4.4 Efficiency Evaluation

We evaluate the efficiency of RPC in terms of token-generation throughput and peak memory usage. All experiments are conducted using an input prompt with 128 tokens and measure throughput for generating sequences of 8192, 16384, 32768 tokens, with a batch size of 16. The compression interval $P$ is set to 4096. Throughput and memory measurements for DeepSeek-R1-Distill-Qwen-7B are obtained on a single NVIDIA H100 SXM GPU, while QwQ-32B evaluations are conducted on four H100 SXM GPUs. Figure 7 presents the throughput and peak memory improvements achieved by RPC relative to the original models with full KV cache. Additional analyses on the efficiency are also provided in Appendix C.3.

**Throughput.** As shown in Figure 7(a), RPC consistently improves token generation throughput with particularly large gains observed for long generation length (e.g., 32768 tokens), a scenario commonly encountered with reasoning LLMs. RPC achieves 1.68× throughput improvement when generating 32768 tokens with DeepSeek-R1-Distill-Qwen-7B, and 1.60× throughput improvement when generating 16384 tokens with QwQ-32B. Notably, QwQ-32B with full KV cache cannot handle reasoning tasks with generation lengths of 32768 tokens as it runs out of memory. However, RPC successfully enables token generation at this length.

**Memory Consumption.** As shown in Figure 7(b), RPC effectively reduces peak memory usage by periodically compressing the KV cache. Since peak memory usage includes contributions from model parameters, intermediate activations, and the KV cache, the reduction in peak memory is not

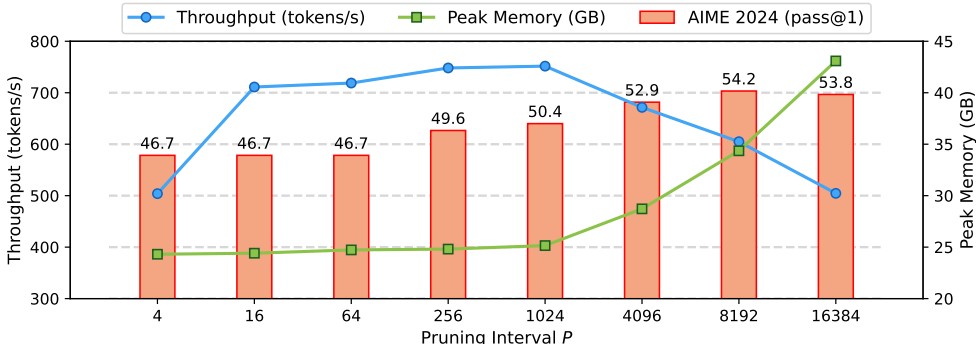

Figure 8: Effect of compression interval $P$ on accuracy, throughput, and peak memory.

directly proportional to the KV cache compression ratio. Nevertheless, as the KV cache becomes the dominant factor in peak memory usage for longer generation lengths, RPC provides increasingly substantial memory savings as generation length grows. For DeepSeek-R1-Distill-Qwen-7B, RPC reduces peak memory usage from 75.7GB to 36.2GB when generating 32768 tokens, thereby RPC achieves over 50% memory reduction. Similarly, for QwQ-32B, RPC reduces the overall memory requirement by over 50%, thereby resolving the out-of-memory issue for the generation of 32768 tokens. These results demonstrate that RPC effectively mitigates the memory bottleneck inherent in long-sequence generation of reasoning LLMs by compressing KV cache.

## 4.5 Ablation Studies

To better understand the effect of key hyperparameters in RPC, we perform ablation studies on DeepSeek-R1-Distill-Qwen-7B using the AIME 2024 dataset. We analyze two critical components: the compression interval $P$, which determines how often KV-cache compression is applied, and the selector window size $R$, which controls the number of recent tokens used for attention-based importance scoring.

**Compression Interval.** We evaluate compression interval $P$ from 4 to 16384 to examine the trade-off between compression interval, reasoning accuracy, and inference efficiency (throughput and peak memory). As shown in Figure 8, reasoning accuracy improves as $P$ increases. It indicates that overly frequent compression can disrupt the reasoning process by prematurely evicting tokens critical for subsequent reasoning steps. However, when $P$ becomes excessively large (e.g., $P = 8192$), throughput declines and peak memory usage rises significantly, as large $P$ delays the KV cache compression. Therefore, selecting an appropriate $P$ value is essential to balance accuracy preservation and efficiency gains. Here, the configurations $P = 4096$ and $P = 1024$ represent practical choices that offer strong balance between performance and efficiency in reasoning-intensive scenarios.

Table 2: Effect of selector window size $R$.

| $P$ | Metric | $R = 1$ | 8 | 32 | 128 |
|------|--------|---------|-----|-----|-----|
| 4096 | AIME 2024 (pass@1) | 49.2 | 48.3 | **52.9** | 49.2 |
| | Throughput (tokens/s) | 662.54 | 662.84 | 671.38 | 673.26 |
| | Peak Memory (GB) | 28.72 | 28.72 | 28.72 | 29.38 |
| 1024 | AIME 2024 (pass@1) | 45.8 | 49.2 | **50.4** | 50.0 |
| | Throughput (tokens/s) | 746.21 | 745.08 | 751.69 | 742.46 |
| | Peak Memory (GB) | 24.62 | 24.62 | 25.15 | 27.37 |

**Selector Window Size.** We evaluate the impact of the selector window size $R$ on the RPC algorithm by evaluating $R \in \{1, 8, 32, 128\}$. As shown in Table 2, small $R$ values such as 1 and 8 yield relatively low accuracy (e.g., below 50% with $P = 4096$), because small $R$ values can result in unstable selection of semantically critical tokens. This effect is more pronounced for $P = 1024$ than $P = 4096$, as tokens are evicted more frequently with smaller $P$. Therefore, $R$ must be sufficiently large to ensure robust importance estimation. However, excessively large $R$ (e.g., 128) can negatively impact accuracy, as older selector tokens may not reflect the current reasoning context effectively. Because varying $R$ has only marginal effects on throughput and peak memory usage, accuracy is

the primary consideration when selecting $R$. Based on our results, $R = 32$ is the best choice as it provides the highest accuracy.

## 5 Conclusion

We introduce Reasoning Path Compression (RPC) for compressing KV cache of reasoning LLMs. We observe that reasoning paths often contain redundant segments and inherent semantic sparsity. RPC leverages this characteristic by periodically compressing the KV cache and employs an importance scoring mechanism based on a selector window composed of recent queries. As RPC does not require any additional training or model modifications, it can be applied to a broad range of reasoning LLMs. Experimental results demonstrate that RPC compresses the KV cache by $4\times$ with accuracy degradation limited to 1.2%. This aggressive KV cache compression results in up to $1.60\times$ throughput improvement. Moreover, RPC successfully resolves the out-of-memory issue encountered by large reasoning models with 32B parameters when generating long reasoning paths of up to 32K tokens, by achieving over 50% reduction of overall memory requirement.

## Acknowledgments and Disclosure of Funding

This work was supported in part by Institute of Information & communications Technology Planning & Evaluation (IITP) grant funded by the Korea government (MSIT) (No.RS-2025-02273157: Development of Low Power Training/Inference Technologies based on AI Semiconductors, RS-2025-10692981: AI Semiconductor Innovation Research Center, RS-2023-00256081: artificial intelligence semiconductor support program to nurture the best talents, No.2021-0-02068: Artificial Intelligence Innovation Hub), Samsung Research Funding Center under Project SRFC-TC1603-53, and BK21 FOUR program at Seoul National University.

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

# A Examples of Redundant Reasoning Paths

In this section, we provide additional redundant reasoning examples from the models that we used for evaluations. These examples clearly illustrate the *semantic sparsity* present in the generation outputs of reasoning LLMs. Ellipses ([...]) indicate omitted portions of the lengthy original output.

In Example A.1, QwQ-32B [12] identifies the correct answer early but redundantly re-derives it multiple times, leading to excessive token generation. Each redundant derivation is highlighted with a box to illustrate the inefficiency. The problem is sampled from AIME 2024 dataset [13].

---

**Example A.1**

**Problem.** There exist real numbers $x$ and $y$, both greater than 1, such that

$$\log_x(y^x) = \log_y(x^{4y}) = 10.$$

Find $xy$.

**QwQ-32B Output.** *<think>* Okay, so I have this problem here: there exist real numbers $x$ and $y$, both greater than 1, such that $\log_x(y^x) = 10$, and $\log_y(x^{4y}) = 10$. I need to find the product $xy$. Hmm, logarithms with variables in the base and the argument. Let me try to unpack this step by step.

First, let's recall some logarithm properties. [...] So now, I have two equations:

$$1. \quad x \cdot a = 10$$

$$2. \quad \frac{4y}{a} = 10$$

So maybe I can solve for $a$ from one equation and substitute into the other?
From equation 1: $a = \frac{10}{x}$
Plugging into equation 2:

$$\frac{4y}{10/x} = 10 \Rightarrow 4y \cdot \frac{x}{10} = 10 \Rightarrow \frac{4xy}{10} = 10 \Rightarrow 4xy = 100 \Rightarrow \boxed{xy = 25}$$

Wait, hold on, that seems too straightforward. But let me check.
Wait, step by step:
Equation 2 after substitution:
$4y$ divided by $\left(\frac{10}{x}\right)$ is equal to 10. So dividing by $\left(\frac{10}{x}\right)$ is multiplying by $\frac{x}{10}$. Therefore,

$$4y \cdot \frac{x}{10} = 10 \Rightarrow \frac{4xy}{10} = 10 \Rightarrow 4xy = 100 \Rightarrow \boxed{xy = 25}$$

Hmm, so is that the answer? Wait, but let me make sure I didn't skip any steps where assumptions might be wrong. Let's check if this holds. [...]
Let's check the original equations.
First equation: $\log_5(5^5)$. Wait, $\log_5(5^5)$ is 5, but the problem says it should be 10. So that's not good. Hmm. So $x = 5$ and $y = 5$ is not a solution. Hmm, so maybe my assumption that $xy = 25$ is correct but $x$ and $y$ individually aren't integers?
Alternatively, maybe I made an error in the substitution. Let me check again. [...]
Let me denote Equation 1 as $\frac{x \cdot v}{u} = 10$ and Equation 2 as $\frac{4y \cdot u}{v} = 10$.
Let me multiply these two equations together:

$$\left(\frac{x \cdot v}{u}\right) \cdot \left(\frac{4y \cdot u}{v}\right) = 10 \cdot 10 \Rightarrow 4xy = 100 \Rightarrow \boxed{xy = 25}$$

Therefore, despite the complication with substitution, the product $xy$ is indeed $\boxed{25}$. However, the problem states that such real numbers $x$ and $y$ exist, so even if we can't find exact values for $x$ and $y$, the product is $\boxed{25}$.
Wait, but maybe we need to confirm that there actually exist such $x$ and $y$? [...]
Therefore, even though we can't get exact values for $x$ and $y$, the product $xy = 25$ is indeed correct, and such numbers exist. Therefore, the answer is $\boxed{25}$.

---

Wait, but the problem says "Find $xy$", so maybe 25 is the answer, and the existence is given, so even without finding $x$ and $y$, through the algebra we can get that $xy = 25$. So maybe the answer is $\boxed{25}$. [...]

$\log_y(x) = \frac{1}{\log_x(y)} \Rightarrow \frac{5}{2y} = \frac{1}{10/x} \Rightarrow \frac{5}{2y} = \frac{x}{10}$

Cross-multiplying: $5 \cdot 10 = 2y \cdot x \Rightarrow 50 = 2xy \Rightarrow \boxed{xy = 25}$

Oh! Wait, this is a much simpler way! I didn't need to go through all the substitutions and transcendental equations. Since $\log_y(x)$ is $\frac{1}{\log_x(y)}$, substituting the expressions from both equations gives directly the relation. So indeed,

$$\frac{5}{2y} = \frac{x}{10} \Rightarrow 50 = 2xy \Rightarrow \boxed{xy = 25}.$$

So that's straightforward. The earlier complication was because I tried to substitute variables and ended up in a loop, but this reciprocal relationship is key here.

Therefore, the answer is $\boxed{25}$. So I think that's solid.
**Final Answer**

$$\boxed{25}$$

*</think>* [...]

A similar pattern is observed in Example A.2, where DeepSeek-R1-Distill-Qwen-7B [3] repeatedly re-computes the correct solution despite having already reached it. Each redundant derivation is highlighted with a box to illustrate the inefficiency. The problem is sampled from GSM8k dataset [34].

---

**Example A.2**

**Problem.** Alice has 20 quarters. She wants to exchange them for nickels and so she goes to the bank. After getting back from the bank, she discovers that 20% of the nickels are iron nickels worth $3 each. What is the total value of her money now?

**DeepSeek-R1-Distill-Qwen-7B Output.** *<think>* Alright, so Alice has 20 quarters, and she wants to exchange them for nickels. Hmm, okay, let me think about this. First, I need to figure out how much money Alice has initially. Quarters are worth 25 cents each, right? So, if she has 20 quarters, I can calculate the total value by multiplying 20 by 25 cents. Let me write that down: 20 quarters * $0.25 per quarter. So, 20 * 0.25 equals... hmm, 20 divided by 4 is 5, so 20 * 0.25 is $5.00. Okay, so Alice has $5.00 in quarters. [...]

So, now I need to calculate the total value of her money now. That would be the value of the iron nickels plus the value of the regular nickels.
First, the iron nickels: 20 nickels * $3 each. Let me compute that: 20 * 3 = $60.00.
Then, the regular nickels: there are 100 total nickels, minus 20 iron nickels, so 80 regular nickels. Each regular nickel is worth $0.05, so 80 * 0.05 = $4.00. So, adding those together, $60 + $4 = $\boxed{\$64.00}$.

Wait, that seems like a big jump from $5.00 to $64.00. Let me double-check my calculations.
She started with 20 quarters, which is 20 * $0.25 = $5.00. Exchanged them all for nickels, so 100 nickels. 20% of 100 is 20, so 20 iron nickels at $3 each is 20 * 3 = $60. The remaining 80 nickels are regular, so 80 * 0.05 = $4. So, total is $60 + $4 = $\boxed{\$64}$. That seems correct. But let me think again about the exchange. [...]

So, 20% of her nickels are iron nickels worth $3 each, so she effectively has 20 coins worth $3 each and 80 coins worth $0.05 each. So, total value is indeed 203 + 800.05 = 60 + 4 = $\boxed{\$64}$.

Wait, but in reality, exchanging quarters for nickels should just be a different form

---

of the same money. So, why is the total value increasing? [...] Wait, maybe I'm overcomplicating. The key is that she exchanges all her quarters into nickels, which would give her 100 nickels, but 20% of them are iron nickels worth $3 each, so 20 nickels at $3 and 80 at $0.05. So, the total value is $20 \cdot 3 + 80 \cdot 0.05 = 60 + 4 = \boxed{\$64}$. So, that's the total value.

But let me think again, is this possible? Because exchanging 20 quarters for nickels should just be a different form of the same money, so why would the total value increase? [...] But in this case, 20 of them are worth $3 each, so total value is $20 \cdot 3 + 80 \cdot 0.05 = 60 + 4 = \boxed{\$64.00}$. So, in effect, she's getting more money by exchanging her quarters into nickels where some are worth more. So, this seems to be the case.

So, maybe the answer is $\boxed{\$64.00}$.

But just to make sure, let me recast the problem.
Total initial value: 20 quarters * $0.25 = $5.00.
After exchanging, she has 100 nickels. But 20% of those are iron nickels worth $3. So, 20 nickels * $3 = $60, and 80 nickels * $0.05 = $4. So, total is $60 + $4 = \boxed{\$64}$.

Yes, that seems correct. Alternatively, maybe the question is trying to trick me into thinking that the total value is still $5, but no, since 20% of the nickels are worth more, so the total value increases.

Therefore, the total value of her money now is $\boxed{\$64.00}$.

Final Answer The total value of her money now is $\boxed{64}$ dollars. *</think>* [...]

In Example A.3, QwQ-32B exhibits a similar pattern to Figure 1, repeatedly stating previously mentioned priors or revisiting earlier examined cases during the problem-solving process. The problem is sampled from GPQA-Diamond dataset [35].

---

**Example A.3**

**Problem.** You perform a high-throughput experiment on white lupine to find genes contributing to resistance to the fungal disease anthracnose. As a result, you receive three candidate genes of unknown function – G1, G2, and G3. You create three knock-out mutants, g1, g2, and g3, and a set of double-mutants, g1g2, g1g3, and g2g3. You know that at least one of these genes is a transcription factor acting upstream of (an)other gene(s). You start to test those mutant plants: do they have a higher sensitivity to anthracnose than the wild-type because they cannot produce certain gene products? After tests with the pathogen, you receive the following results where 100% is the level of resistance to the pathogen in control; 50% is half of the control's resistance; 25% is a quarter of the control's resistance; 0% – all plants show signs of infection:

- resistance of g1: 75% of control
- resistance of g2: 0% from control
- resistance of g3: 50% from control
- resistance of g1g3: 10% from control
- resistance of g2g3: 0% from control
- resistance of g1g2: 0% from control

Which conclusion regarding those genes' interaction can you draw from this experiment?

**QwQ-32B Output.** *<think>* Okay, so I have this problem about white lupine genes and their resistance to anthracnose. Let me try to figure this out step by step.
First, the setup: There are three genes, G1, G2, G3. They made knockout mutants for each (g1, g2, g3), and all the double mutants (g1g2, g1g3, g2g3). The question is about their interactions, especially since at least one is a transcription factor upstream of others. The resistance levels are given for each mutant and double mutant. The resistance percentages are as follows:

---

- g1: 75%
- g2: 0%
- g3: 50%
- g1g3: 10%
- g2g3: 0%
- g1g2: 0%

Hmm. Let me think about how these genes might interact. Since a transcription factor is upstream, it probably controls the expression of another gene. If a gene is a transcription factor, knocking it out would prevent the downstream gene from being expressed, leading to a loss of function. Looking at the single mutants first:

- g2 has 0% resistance. That's the lowest, so losing G2 completely breaks resistance.
- g1 is 75%, so it's still somewhat resistant.
- g3 is 50%, so halfway.

Now, the double mutants:

- g1g2 is 0%: same as g2 alone.
- g2g3 is also 0%: same as g2.
- g1g3 is 10%: which is worse than either g1 (75%) or g3 (50%).

Wait, so when you knock out G1 and G3 together, the resistance drops further to 10%, which is worse than either single knockout. That suggests that G1 and G3 might be acting in parallel pathways? Because their combined loss is worse than either alone.

But then looking at g2g3 and g1g2, both are 0%, same as g2. So maybe G2 is epistatic over the others? Because when you have g2, it's already 0%, so adding another mutation doesn't make it worse. That might mean that G2 is the main gene, and the others are downstream or in a different pathway that doesn't matter once G2 is gone.

Alternatively, maybe G2 is the transcription factor, and it's required for the expression of G1 and/or G3. If G2 is a transcription factor, then knocking it out would prevent expression of the downstream genes. So if G2 is upstream of G1 and G3, then in the g2 mutant, both G1 and G3 wouldn't be expressed either. But then why does the g1 mutant still have 75% resistance? That seems contradictory.

Wait, maybe G2 is not the transcription factor here. Let me think again.
Alternatively, maybe G3 is the transcription factor. Let's see. If G3 is a transcription factor that controls G1 and/or G2. Then, in the g3 mutant, both G1 and G2 might be downregulated. But then the g3 single mutant has 50% resistance. The g2 mutant has 0
Alternatively, G1 might be a transcription factor. Let's consider that. If G1 is a transcription factor, and it's controlling G2 or G3.

Looking at the double mutants:
g1g3 has 10%. Since G1 alone is 75%, and G3 alone is 50%, combining them gives 10%, which is a significant drop. So maybe G1 and G3 are in parallel pathways, and their combined loss is more severe. [...]
Alternatively, maybe G2 is a transcription factor that's needed to activate G1 and G3. Let me think through that. If G2 is a transcription factor, then in the g2 mutant, neither G1 nor G3 would be expressed. But then why does the g1 mutant still have 75% resistance? If G1 is not expressed in the g2 mutant, but the g1 mutant (which still has G2) has 75%, that suggests that G1 can function even if G2 is present. Wait, this is getting confusing.

Alternatively, perhaps G3 is a transcription factor that activates G2. Let's see: If G3 is a transcription factor for G2, then in the g3 mutant, G2 might not be expressed. Then,

the g3 single would have no G2, so resistance would be 0%, but in reality g3 is 50%, so that can't be right.
Hmm. Let's try another approach. [...]

# B Redundancy Reduction

## B.1 Details of Embedding-based Redundancy Measurements

This section provides details of the embedding-based analysis described in Section 4.2 used to demonstrate the redundancy-reducing effect of RPC.

For this analysis, we generated outputs for each sample in the AIME 2024 dataset [13] using both the full KV baseline and RPC. Generation was stopped after the $4128^{\text{th}}$ decoding step, where the first compression of RPC is completed, given $P = 4096$ and $R = 32$. The generated outputs are separated into individual sentences, and each sentence is embedded using the all-MiniLM-L6-v2 model with Sentence Transformers library [36]. The model maps each token sequence to a 384-dimensional vector.

Pairwise cosine similarities were computed between all sentence embeddings within the same generated output. Two sentences with cosine similarity above 0.75 were considered semantically similar. Finally, we defined the *redundancy rate* as the proportion of sentences that have more than $N$ semantically similar counterparts within the same output, where $N \in 1, 2, 4$.

## B.2 Visualized Examples of Token Selection with RPC

In this section, we provide qualitative examples illustrating how RPC selects tokens to be retained in the KV cache.

In the following examples, token colors indicate how frequently each token is retained across layers: tokens preserved by a larger proportion of layers appear in warmer colors, following the scale defined in the color legend below.

**Legend:**

- Black: Unselected or low selection (<40%)
- Yellow: Medium selection (40-60%)
- Orange: High selection (60-80%)
- Red: Very high selection ($\geq$80%)

Examples B.1 and B.2 are sampled from outputs of DeepSeek-R1-Distill-Qwen-7B [3] and QwQ-32B [12], respectively, on the AIME 2024 dataset.

While it is natural that recent tokens tend to exhibit higher retention frequencies due to LLMs' tendency to assign larger attention weights to local tokens, the selection pattern from a semantic perspective can be broadly categorized into two types:

(1) **Information-bearing tokens:** Tokens containing critical information that advances the reasoning process, such as conclusions of intermediate steps.

(2) **Transition tokens:** Tokens indicating shifts or boundaries between reasoning stages, which, although not semantically rich on their own, play an important contextual role through the attention mechanism by aligning and integrating information across reasoning steps.

This qualitative analysis demonstrates that RPC effectively preserves essential reasoning content while filtering out redundant or outdated derivations, thereby maintaining the continuity and integrity of the reasoning trajectory.

---

**Example B.1**

**Problem.** Let $\triangle ABC$ have circumcenter $O$ and incenter $I$ with $\overline{IA} \perp \overline{OI}$, circumradius $R = 13$, and inradius $r = 6$. Find $AB \cdot AC$.

**DeepSeek-R1-Qwen-7B Output.** *<think>* Okay, so I need to find the product AB times AC in triangle ABC, given that the circumradius is 13, the inradius is 6, and the segment IA is perpendicular to OI, where O is the circumcenter and I is the incenter. Hmm,

---

that seems a bit involved, but let me try to break it down step by step.

First, let me recall some properties and formulas related to triangle centers, circumradius, inradius, and maybe some coordinate geometry. Since the problem involves perpendicularity between IA and OI, maybe coordinate geometry could be a good approach here. Alternatively, there might be some formula that connects these elements directly. Let me think.

I know that in any triangle, the distance between the incenter and circumcenter can be expressed by the formula: $OI^2 = R(R - 2r)$ where R is the circumradius and r is the inradius.

Wait, is that correct? Let me verify. Hmm, actually, I think the formula is $OI^2 = R^2 - 2Rr$. Yes, that's right. So OI squared equals R squared minus 2 times R times r. So substituting the given values, R is 13 and r is 6, so $OI^2 = 13^2 - 2*13*6 = 169 - 156 = 13$. Therefore, OI is sqrt(13). Hmm, but the problem states that IA is perpendicular to OI. Maybe this can help relate IA and OI? [...]

The inradius r is related to the area and semiperimeter: $r = area / s$, where s = (a + b + c)/2. The circumradius R is related to the sides by the formula a = 2R sin A, and similar for other sides. Also, the distance between O and I is given by $OI^2 = R^2 - 2Rr$, which we already computed as 13. So OI = sqrt(13). But we also have that IA is perpendicular to OI. So maybe we can use vectors or coordinate geometry to relate these.

Alternatively, perhaps using the formula for the distance from the incenter to a vertex. The length IA can be expressed in terms of the angles and inradius. Let me recall that in a triangle, the distance from the incenter to vertex A is given by: $IA = \sqrt{r^2 + \left( s - a \right)^2 }$

Wait, is that correct? Let me see. The formula for the distance from the incenter to vertex A: yes, in triangle ABC, the distance from I to A is sqrt(r² + (s - a)²), where s is the semiperimeter and a is the side BC. Similarly for other vertices. [...]

But since A is a vertex of the triangle on the circumradius of 13, so OA = 13, so $x_A^2 + y_A^2 = 13^2 = 169$.

Hmm, this might get complicated with too many variables. Maybe there is a better way.

Alternatively, since O is the circumcenter, coordinates can be set such that triangle ABC is inscribed in a circle of radius 13. Let me perhaps place point A at (13, 0) for simplicity, so coordinates of A are (13, 0). Then O is at (0, 0). Then points B and C are somewhere on the circle $x^2 + y^2 = 169$. [...]

Expressed as vectors, OI is (I - O) and IA is (A - I). Since O is the origin, OI is just the coordinates of I, and IA is (A - I). So their dot product is I • (A - I) = 0.

Therefore, $I • A - |I|^2 = 0.$

Therefore, $I • A = |I|^2$. Hmm, that's an equation relating the coordinates of I and A. Since A is on the circumcircle, |A| = R = 13. [...]

Therefore, $IA^2 = 169 - 13 = 156$, so IA = sqrt(156) = 2*sqrt(39). Hmm, that seems useful.

So IA is sqrt(156). But we also have IA = r / sin(α/2). Since r = 6, so:sqrt(156) = 6 / sin(α/2)

Therefore, $\sin(\alpha/2)$ = 6 / sqrt(156) = 6 / (2*sqrt(39)) ) = 3 / sqrt(39) = sqrt(39)/13. Wait, let's compute that again:

Wait 6 divided by sqrt(156) is equal to 6 divided by (sqrt(4*39)) = 6/(2*sqrt(39)) = 3/sqrt(39). Rationalizing the denominator: 3/sqrt(39) = 3*sqrt(39)/39 = sqrt(39)/13. So $\sin(\alpha/2)$ = sqrt(39)/13.

Therefore, $\sin(\alpha/2)$ = sqrt(39)/13. Let me note that sqrt(39)/13 is equal to sqrt(39)/13, which is approximately 6.245/13 ≈ 0.48, which is plausible.

Then, $\cos(\alpha/2)$ would be sqrt(1 - (39/169)) = sqrt(130/169) = sqrt(130)/13.

Therefore, $\cos(\alpha/2)$ = sqrt(130)/13.

Then, $\sin \alpha$ = 2 $\sin(\alpha/2) \cos(\alpha/2)$ = 2*(sqrt(39)/13)*(sqrt(130)/13) = 2*sqrt(39*130)/($13^2$).

Compute 39*130: 39*100 = 3900, 39*30 = 1170, so total 3900 + 1170 = 5070. So sqrt(5070).

Hmm, 5070 factors: 5070 = 10*507 = 10*3*13*13. Wait, 507 divided by 13 is 39, so 507 = 13*39 = 13*3*13.

Wait, 39 is 3*13, so 5070 = 10*3*13*13. Therefore sqrt(5070) = sqrt(10*3*$13^2$) ) = 13*sqrt(30). Because 10*3 = 30. So sqrt(5070) = 13*sqrt(30).

Therefore, $\sin \alpha$ = 2*(13*sqrt(30))/($13^2$) ) = (2 sqrt(30))/13.

Thus $\sin \alpha$ = (2 sqrt(30))/13.

Hmm, so angle A has sine equal to $2\sqrt{30}/13$.

Therefore, the side BC can be found via the Law of Sines:
a = BC = 2R $\sin \alpha$ = 2*13*(2$\sqrt{30}$/13) = 4$\sqrt{30}$. [...]

---

**Example B.2**

**Problem.** Let $A$, $B$, $C$, and $D$ be points on the hyperbola $\dfrac{x^2}{20} - \dfrac{y^2}{24} = 1$ such that $ABCD$ is a rhombus whose diagonals intersect at the origin. Find the greatest real number that is less than $BD^2$ for all such rhombi.

**QwQ-32B Output.** *<think>* Okay, so I have this problem here about a hyperbola and a rhombus inscribed in it. The hyperbola equation is given as x²/20 - y²/24 = 1. The rhombus ABCD has its diagonals intersecting at the origin, and I need to find the greatest real number that is less than BD² for all such rhombi. Hmm, okay, let me break this down step by step. [...]

But first, let me think about coordinates. Let me denote points A and C as (x1, y1) and (-x1, -y1), and points B and D as (x2, y2) and (-x2, -y2). Since all four points lie on the hyperbola, they must satisfy the equation x²/20 - y²/24 = 1. So for points A and C, (x1)²/20 - (y1)²/24 = 1, and similarly for points B and D, (x2)²/22 - (y2)²/24 = 1. [...]

Let me consider parametrizing the hyperbola. The hyperbola equation is x²/20 - y²/24 =1. For hyperbola parametrization, we can use hyperbolic functions. Let me recall that a standard hyperbola x²/a² - y²/b² =1 can be parametrized as (a sec$\theta$, b tan$\theta$), or using hyperbola parameters. Alternatively, using coordinates:

Let me set x = sqrt(20) sec$\theta$, y = sqrt(24) tan$\theta$. That would satisfy the hyperbola equation. Alternatively, perhaps using coordinates in terms of parameters t and s for points A and B.

Alternatively, since the hyperbola is symmetric, maybe we can set point A as (x, y) and point B as (u, v), such that x²/20 - y²/24 =1 and u²/20 - v²/24 =1. Then, the condition is x u + y v =0 (from x1x2 + y1y2 =0). Then, BD² would be the squared distance between points B and D. Since D is (-u, -v), BD is the distance between (u, v) and (-u, -v), which is sqrt[(2u)^2 + (2v)^2] so BD squared is (2u)^2 + (2v)^2 =4(u² + v²). [...]

Alternatively, maybe use Lagrange multipliers to minimize u² + v² subject to the constraints:

1. u²/20 - v²/24 =1,
2. x u + y v =0,

and also x²/20 - y²/24 =1.

But this might get complicated with multiple variables. Let me see if I can express variables in terms of each other.

Let me suppose that OA is (x,y) and OB is (u,v). Since OA and OB are perpendicular, then u = k y, v = -k x for some scalar k. Wait, because if (x,y) and (u,v) are perpendicular, then (u,v) is a scalar multiple of (-y, x) or (y, -x). Let me check:

If OA is (x,y), then a perpendicular vector would be (-y, x) or (y, -x). So, (u,v) must be a scalar multiple of that. Let me suppose (u,v) = $\lambda$*(-y, x). Then, substituting into the hyperbola equation for B:

u²/20 - v²/24 = ( ($\lambda$² y²)/20 ) - ( ($\lambda$² x²)/24 ) = $\lambda$² ( y²/20 - x²/24 ) =1.

Wait, but the hyperbola equation for point A is x²/20 - y²/24 =1, so y²/24 = (x²/20 -1). Hmm, so let me rearrange that.

Wait, let me see:

Given (u,v) = $\lambda$*(-y, x). Then, substituting into the hyperbola equation:

u²/20 - v²/24 = ($\lambda$² y²)/20 - ($\lambda$² x²)/24 = $\lambda$² ( y²/20 - x²/24 ) =1.

But from point A's equation, x²/20 - y²/24 =1, which can be rearranged to y²/24 = (x²/20 -1). Therefore, y² =24*(x²/20 -1). Let me plug that into the expression above.

So, y²/20 - x²/24 = (24*(x²/20 -1))/20 - x²/24.

Wait, let me compute that step by step.

First, compute y²/20:

y²/20 = [24*(x²/20 -1)] /20 = 24/(20)*(x²/20 -1) = (24/20)*(x²/20 -1)

Wait, perhaps this is getting too convoluted. Let me think differently.

Let me denote S = u² + v². Since BD squared is 4S, we need to find the minimal value of S given that (u,v) lies on the hyperbola and (u,v) is perpendicular to (x,y) which is also on the hyperbola.

Alternatively, since (u,v) is along the direction perpendicular to (x,y), let me parametrise

(u,v) as a scalar multiple of (-y, x). So let me write $u = -\lambda y$, $v = \lambda x$. Then, substituting into the hyperbola equation for B:

$u^2/20 - v^2/24 = (\lambda^2 y^2)/20 - (\lambda^2 x^2)/24 = \lambda^2 ( y^2/20 - x^2/24 ) =1$.

From point A's hyperbola equation, $x^2/20 - y^2/24 =1$, so rearranged: $y^2/24 = x^2/20 -1$, so $y^2 =24*(x^2/20 -1)$. Let me substitute that into the expression above.

First, compute $y^2/20 - x^2/24$:

$y^2/20 = [24*(x^2/20 -1)] /20 = (24/20)*(x^2/20 -1) = (6/5)*(x^2/20 -1)$

$x^2/24$ is just $x^2/24$.

Therefore, $y^2/20 - x^2/24 = (6/5)(x^2/20 -1) - x^2/24$

Let me compute this term:

First, $(6/5)(x^2/20 -1) = (6x^2)/(100) - 6/5 = (3x^2)/50 - 6/5$

Then subtract $x^2/24$:

$(3x^2/50 - 6/5) - x^2/24 = (3x^2/50 - x^2/24) -6/5$ [...]

# C   Extra Experimental Results

## C.1   Output Length Statistics across Evaluation Datasets

Table 3 summarizes the output length statistics of DeepSeek-R1-Distill-Qwen-7B [3] and QwQ-32B [12], measured under the full KV cache setting. For both models, AIME 2024 [13] and LiveCodeBench [25] exhibit long reasoning outputs (over 10K tokens on average), whereas IFEval produces much shorter outputs. Accordingly, compression in RPC is triggered multiple times for long-context benchmarks but rarely for short ones, which justifies using a consistent $P$ value (1024 or 4096) across all datasets to prevent excessive KV cache growth.

Table 3: Output length distribution under full KV cache setting.

| Dataset | DeepSeek-R1-Distill-Qwen-7B | | | | QwQ-32B | | | |
|---|---|---|---|---|---|---|---|---|
| | Mean | Min | Max | Std | Mean | Min | Max | Std |
| AIME 2024 | 13668.6 | 2413 | 32768 | 9356.4 | 13834.6 | 2747 | 32768 | 7365.5 |
| LiveCodeBench | 11889.1 | 809 | 32768 | 7066.2 | 13454.6 | 491 | 32768 | 9692.1 |
| IFEval | 1778.4 | 190 | 32768 | 5073.3 | 1336.9 | 144 | 32768 | 1844.3 |

## C.2   Granularity of Attention Score Aggregation for Token Selection

To examine how the granularity of attention score aggregation for token selection impacts accuracy, we compare three aggregation schemes: *layer-wise*, *group-wise* (key–value group), and *head-wise* (no aggregation). In our approach, attention scores are aggregated at the layer level, meaning that saliency is averaged across all heads within each layer.

Table 4 presents results for DeepSeek-R1-Distill-Qwen-7B on AIME 2024 dataset under different aggregation granularities. The results show that layer-wise aggregation consistently yields the highest accuracy, indicating that averaging attention scores across heads helps stabilize the estimation of token importance and preserve overall performance after compression.

Table 4: AIME 2024 (pass@1) results for DeepSeek-R1-Distill-Qwen-7B with different attention score aggregation granularities.

| $P$ | Layer Aggregation | Group Aggregation | Head (No Aggregation) |
|---|---|---|---|
| 4096 | 52.9 | 50.8 | 49.6 |
| 1024 | 50.4 | 50.4 | 47.5 |

Layer-wise aggregation offers a coarser yet more reliable estimation of token saliency than head-level aggregation, which often suffers from high variance and instability across heads. This observation is consistent with previous work. TOVA [11] reported that layer-level token selection outperformed head-level selection in terms of perplexity.

Moreover, in models using grouped-query attention (GQA) [37], multiple attention heads share a single KV head. Performing token selection separately for each head in such architectures would require maintaining distinct KV caches per head, introducing significant memory overhead and defeating the purpose of compression. Therefore, even aside from accuracy, head-level selection is impractical for GQA-based models.

Overall, the layer-level token selection strategy adopted in RPC offers a practical and stable solution for compressing the KV cache of reasoning LLMs.

## C.3   Efficiency Evaluation

We evaluate the efficiency gains of RPC by comparing decoding throughput and peak memory usage against the inference with full KV cache. Specifically, we report results for the default $4\times$ compression setting of RPC with two compression intervals, $P = 1024$ and $P = 4096$. Throughput is measured in tokens per second, and peak memory reflects the maximum GPU memory consumption during generation.

Measurements were conducted using two reasoning models: DeepSeek-R1-Distill-Qwen-7B and QwQ-32B. For DeepSeek-R1-Distill-Qwen-7B, evaluations were performed on a single NVIDIA H100 SXM GPU, while QwQ-32B was tested using 4 NVIDIA H100 SXM GPUs in parallel. We fix the input length to 128 tokens and vary the generation length across 4096, 8192, 16384, and 32768 tokens. Batch size is varied across 8, 16, and 32 to assess scalability under different workloads.

Results for DeepSeek-R1-Distill-Qwen-7B are shown in Table 5, and the corresponding results for QwQ-32B are reported in Table 6.

Table 5: DeepSeek-R1-Distill-Qwen-7B's throughput and peak memory usage by batch size and generation length.

| Metric | Batch Size | 4096 | 8192 | 16384 | 32768 |
|---|---|---|---|---|---|
| **Full KV Cache** | | | | | |
| Throughput (tokens/s) | 8 | 401.50 | 368.72 | 330.41 | 256.92 |
| | 16 | 669.53 | 653.04 | 504.50 | 342.88 |
| | 32 | 1328.58 | 1031.51 | 671.83 | OOM |
| Peak Memory (GB) | 8 | 19.20 | 22.95 | 30.47 | 45.50 |
| | 16 | 23.09 | 30.60 | 45.63 | 75.70 |
| | 32 | 30.86 | 45.89 | 75.96 | OOM |
| **RPC** ($P = 1024$) | | | | | |
| Throughput (tokens/s) | 8 | 448.19 | 428.31 | 407.00 | 385.00 |
| | 16 | 848.75 | 794.62 | 751.69 | 650.20 |
| | 32 | 1504.40 | 1499.80 | 1288.51 | 977.11 |
| Peak Memory (GB) | 8 | 17.08 | 18.02 | 20.27 | 24.75 |
| | 16 | 18.86 | 20.74 | 25.15 | 34.20 |
| | 32 | 22.40 | 26.16 | 35.00 | 53.08 |
| **RPC** ($P = 4096$) | | | | | |
| Throughput (tokens/s) | 8 | 406.43 | 420.62 | 385.75 | 362.75 |
| | 16 | 753.11 | 708.95 | 671.38 | 575.21 |
| | 32 | 1318.33 | 1247.44 | 1064.05 | 883.43 |
| Peak Memory (GB) | 8 | 19.20 | 20.14 | 22.02 | 25.77 |
| | 16 | 23.09 | 24.96 | 28.72 | 36.24 |
| | 32 | 30.86 | 34.62 | 42.13 | 57.16 |

The results show that RPC consistently improves decoding efficiency over full KV cache inference across various batch sizes and generation lengths. As the batch size increases, both the throughput gains and peak memory reductions become more pronounced. This is because larger batches amplify the memory bottleneck imposed by the growing KV cache, allowing RPC's compression to better utilize available GPU compute resources. Notably, full KV cache inference results in out-of-memory (OOM) errors for DeepSeek-R1-Distill-Qwen-7B when the batch size is 32 and the generation length reaches 32768, and for QwQ-32B when the batch size is 16 at 32768 tokens or 32 at 16384 tokens or longer. In contrast, RPC enables successful generation under all of these settings.

When comparing compression intervals, $P = 1024$ achieves slightly higher throughput and lower peak memory than $P = 4096$ across both models. While $P = 1024$ offers stronger compression, it may come at a modest accuracy cost, as shown in Section 4.3. Therefore, $P = 1024$ and $P = 4096$ can be considered complementary settings: the former prioritizes efficiency, and the latter provides a more balanced trade-off between performance and accuracy.

Table 6: QwQ-32B's throughput and peak memory usage by batch size and generation length.

| Metric | Batch Size | 4096 | 8192 | 16384 | 32768 |
|---|---|---|---|---|---|
| **Full KV Cache** | | | | | |
| Throughput (tokens/s) | 8 | 128.79 | 109.80 | 93.28 | 64.85 |
| | 16 | 213.75 | 173.99 | 117.51 | OOM |
| | 32 | 351.34 | 228.59 | OOM | OOM |
| Peak Memory (GB) | 8 | 83.40 | 100.58 | 134.94 | 203.66 |
| | 16 | 101.14 | 135.50 | 204.22 | OOM |
| | 32 | 136.61 | 205.33 | OOM | OOM |
| **RPC** ($P = 1024$) | | | | | |
| Throughput (tokens/s) | 8 | 135.79 | 131.97 | 124.45 | 111.84 |
| | 16 | 238.76 | 229.22 | 176.73 | 178.06 |
| | 32 | 411.42 | 392.04 | 328.56 | 246.81 |
| Peak Memory (GB) | 8 | 73.75 | 78.42 | 89.19 | 111.84 |
| | 16 | 81.81 | 91.24 | 112.77 | 155.81 |
| | 32 | 97.95 | 116.70 | 159.78 | 245.94 |
| **RPC** ($P = 4096$) | | | | | |
| Throughput (tokens/s) | 8 | 126.59 | 113.32 | 115.75 | 102.57 |
| | 16 | 214.27 | 207.28 | 187.97 | 147.26 |
| | 32 | 345.02 | 314.67 | 279.34 | 208.48 |
| Peak Memory (GB) | 8 | 83.40 | 87.70 | 96.28 | 114.53 |
| | 16 | 101.14 | 109.73 | 126.90 | 163.40 |
| | 32 | 136.61 | 153.79 | 188.14 | 261.13 |

### C.4 Effect of Aggressive Compression

To assess the robustness of RPC under extreme compression, we evaluate its performance with a target compression ratio of $8\times$. This setting represents a highly aggressive compression scenario where only one-eighth of the generated tokens' KV entries are retained over time. Table 7 shows the resulting performance across the three benchmark datasets.

Table 7: Accuracy (%) of RPC $8\times$ compared to RPC $4\times$ and full KV cache.

| Method | DeepSeek-R1-Distill-Qwen-7B | | | QwQ-32B | | |
|---|---|---|---|---|---|---|
| | AIME 2024 (pass@1) | LiveCodeBench (pass@1) | IFEval (pass@1) | AIME 2024 (pass@1) | LiveCodeBench (pass@1) | IFEval (pass@1) |
| Full KV Cache | 55.5 | 37.6 | 55.1 | 79.5 | 63.4 | 83.9 |
| **RPC 4× Best** | 52.9 | 35.9 | 57.3 | 78.3 | 62.2 | 82.6 |
| **RPC 8× ($P = 4096$)** | 47.5 | 32.8 | 55.1 | 72.1 | 57.2 | 84.3 |
| **RPC 8× ($P = 1024$)** | 37.5 | 27.2 | 58.4 | 72.1 | 57.4 | 82.8 |

Both models exhibit a notable performance drop on AIME 2024 and LiveCodeBench under $8\times$ compression, compared to the default $4\times$ setting, indicating the difficulty of preserving reasoning fidelity under extreme compression. Nevertheless, the stronger reasoning model QwQ-32B demonstrates greater robustness, maintaining pass@1 scores close to the results of RPC $4\times$ across both benchmarks. In contrast, on IFEval, a benchmark characterized by lower reasoning difficulty, the performance remains stable or even improves slightly for both models, suggesting that light-weight instruction-following tasks are less sensitive to aggressive KV cache compression.

