# OpenReview forum: "Reasoning Path Compression: Compressing Generation Trajectories for Efficient LLM Reasoning"
_NeurIPS.cc/2025/Conference — NeurIPS 2025 poster_

### Official Review · Reviewer_JM3v · 2025-06-07

**Clarity:** 3
**Significance:** 2
**Originality:** 3
**Rating:** 4
**Confidence:** 3

**Summary:**

This work proposes a novel technique for inference acceleration, termed Reasoning Path Compression (RPC), which requires no model training. The method is designed to enhance generation efficiency by exploiting the inherent semantic sparsity of reasoning paths. Specifically, RPC evaluates and quantifies the importance of KV cache entries through a selector window comprising the latest queries. Based on this scoring, the method performs periodic compression of the KV cache, preserving only high-importance entries. Experimental results demonstrate that applying RPC increases the generation throughput of the Qwen-32B model by up to 1.60× over the baseline (full KV cache), with the accuracy drop on the AIME 2024 benchmark contained to 1.2%.

**Questions:**

It appears the throughput experiments were conducted with fixed prompt and decode lengths. I suggest adding batch throughput experiments using realistic reasoning benchmarks to strengthen the evaluation.

I suggest providing a comparative analysis between the selected kv cache and the actual generated reasoning tokens to substantiate the claim that the algorithm effectively eliminates redundant reasoning logic.

I recommend revising the figure 4 to explicitly illustrate how to select tokens to evict.

**Ethical Concerns:**

["NO or VERY MINOR ethics concerns only"]

**Final Justification:**

I thank the authors for their detailed rebuttal. Their responses have addressed the questions I raised regarding the clarity of the methodology and the experimental setup. The new explanations have clarified my previous misunderstandings.

Resolved Issues: My primary concerns about the clarity of the core methodology have been fully resolved by the authors' clarifications.

Remaining Issues: None.

Justification for Score Change: The authors' clear and effective response has directly improved my understanding of the paper's contributions. Therefore, I have raised the score for "Clarity" by one point and will increase my overall score by one point. I believe the paper is now in a stronger position for acceptance.

**Limitations:**

yes

**Paper Formatting Concerns:**

no concern

**Quality:**

2

**Strengths And Weaknesses:**

Strengths:
The paper proposes a novel method for KV cache selection that demonstrates a minimal loss in accuracy on benchmarks such as AIME, LiveCodeBench, and IFEval. Notably, the method significantly reduces memory usage while simultaneously improving throughput. This algorithm is easy to implement and provides decent results.

Weaknesses:
The connection between the proposed motivation and the algorithm appears to be weak.
The decision to permanently discard tokens from the cache, making them unavailable for subsequent decoding steps, seems problematic. This approach could lead to a degradation in accuracy if tokens that are initially deemed unimportant become critical later in the generation process.
Regarding Figure 4, I found it difficult to understand the compression step.

---

> ### Author Rebuttal · Authors · 2025-07-31
>
> **[Response JM3v-1] Weakness 1: Concerns on permanent KV cache eviction**
>
> Thank you for the valuable comment. First, we would like to clarify that there are two major research directions for addressing the **long KV cache problem** in LLM inference:
> 1. **Memory Offloading Approaches** (e.g., QUEST [1]): These methods aim to distribute memory requirement by offloading KV cache entries to host memory and selectively retrieving them during decoding, rather than discarding them permanently. While this strategy has the advantage of fully preserving all KV information, it does **not reduce total memory usage**, and the repeated indexing and retrieval introduce **non-trivial overhead** at each decoding step.
> 2. **Permanent Eviction Approaches** (e.g., TOVA [2], H2O [3]): This direction focuses on permanently evicting KV cache entries, **directly reducing memory usage and avoiding the runtime overhead**. Due to these efficiency benefits, there has been growing interest in designing eviction-based compression methods [2, 3]. As you rightly pointed out, naive eviction strategies can lead to significant accuracy degradation. Therefore, the key challenge in this direction is to design effective compression mechanisms that accurately identify and retain the most critical KV entries to preserve model performance.
>
> Our proposed RPC falls into the second category and we proposed a novel KV cache compression mechanism specialized for reasoning LLMs. The proposed RPC addresses the redundancy in generated reasoning paths through periodic, attention-based eviction of KV cache entries for generated tokens. As shown in Table 1 of our manuscript, RPC achieves better accuracy preservation compared to prior works. This experimental result validates its effectiveness in compressing long reasoning paths.
> We will revise the manuscript to better highlight these distinctions and clarify the rationale behind the permanent eviction strategy in RPC.
>
> Thank you again for your thoughtful feedback.
>
>
> **[Response JM3v-2] Weakness 2 and Question 3: Clarification of Figure 4**
>
> Thank you for your comment. We will revise **Figure 4** to more clearly illustrate the token eviction process in the proposed RPC method. As figure updates are not allowed during the rebuttal period, we outline below our planned revisions to be included in the updated manuscript.
>
> We will modify the figure to visually depict the following key components of RPC:
> 1. **selector window** and its role in evaluating token importance
> 2. **computation of importance scores**
> 3. **selection of tokens to retain or evict** at each compression step.
>
> More specifically, we will clarify the eviction mechanism as follows:
>
>  At each compression interval, RPC identifies which tokens to retain based on attention scores computed between the most recent R token (i.e., the selector window) and all previously generated tokens in the KV cache. To support efficient attention score computation, the query vectors of the selector tokens are cached during generation. Using these cached queries, RPC computes aggregated attention scores across attention heads in each layer to estimate the importance of tokens in the current KV cache.
>
> Importantly, eviction is not limited to recently generated tokens. Instead, RPC considers the entire KV cache, which includes both tokens retained from previous compression steps and newly generated tokens. Tokens are ranked by their aggregated attention scores, and RPC retains only the top-k tokens, along with the R selector tokens. For the N-th compression cycle, k is defined as NP/c, where P is the compression interval and c is the compression ratio. This design allows RPC to naturally phase out outdated and less relevant tokens as reasoning progresses.
>
> **[Response JM3v-3] Question 1: Throughput measurement on real reasoning dataset**
>
> To evaluate throughput under realistic conditions, we measured the throughput of reasoning LLMs on the AIME 2024 benchmark across various batch sizes. As in our original manuscript, R1-Distill-Qwen-7B was evaluated on a single NVIDIA H100 GPU, while QwQ-32B was run on four H100 GPUs. Please note that, since we used a real-world benchmark, the output lengths of samples within a batch vary.
>
> As shown in **Table JM3v-1  JM3v-2**, RPC consistently achieved significant throughput improvements compared to Full KV cache. For example, at batch size 16, RPC achieved up to 2.58× higher throughput than Full KV. At batch size 32, Full KV encountered out-of-memory errors, while RPC successfully completed generation with high throughput and no memory issues.
>
> **Table JM3v-1 Throughput (tokens/sec) measurement on AIME 2024  (R1-Distill-Qwen-7B)**
> | Batch Size | 8      | 16      | 32      |
> |------------|--------|---------|---------|
> | Full KV | 152.92 | 252.12 | OOM|
> | P=4096  | 257.66 | 474.65  | 706.01  |
> | P=1024   | 348.82 | 591.28  | 938.85  |
>
> **Table JM3v-2 Throughput (tokens/sec) measurement on AIME 2024 (QwQ-32B)**
> | Batch Size | 8  | 16 | 32 |
> |------------|-------|--------|--------|
> | Full KV    | 30.10 | OOM    | OOM    |
> | P=4096     | 70.51 | 98.97  | 152.40 |
> | P=1024     | 81.76 | 120.83 | 205.60 |
>
> These results confirm that the efficiency gains of RPC generalize to realistic batched inference on actual reasoning benchmarks, not just under fixed-length scenarios.
>
> **[Response JM3v– 4] Question 2: Analysis on selected KV cache and elimination of redundancy**
>
> Thank you for highlighting the importance of an analysis on the selected KV cache. We fully agree that such analysis is essential to substantiate our claim.
>
> Through observations of the selected KV cache on real examples, we find that the tokens most frequently retained by RPC generally fall into two key categories:
>
> - Tokens containing critical information that advances the reasoning process, such as conclusions of intermediate steps.
> - Tokens indicating transitions between reasoning steps, which, although not semantically rich on their own, play an important contextual role when processed through the attention mechanism, as they help align and integrate information across reasoning stages.
>
> However, due to space constraints, it is difficult to present a full example of the selected KV cache from the long generated outputs. Therefore, we instead conducted a quantitative analysis using embedding-based sentence similarity.
>
> Specifically, we applied sentence-level vector embeddings to the generated outputs and computed pairwise cosine similarity between sentence embeddings. Two sentences were considered “semantically similar” if their cosine similarity exceeded 0.75. Based on this, we defined the redundancy rate as the proportion of sentences that have more than N(N=1,2,4) semantically similar sentences within the same output.
> This analysis was conducted on outputs from three reasoning LLMs (R1-Distill-Qwen-7B, R1-Distill-Llama-8B, and QwQ-32B) using the AIME 2024 dataset.
>
> As shown in **Table JM3v-3**, our analysis reveals that the redundancy rate dropped significantly after applying RPC. This result provides strong evidence that RPC successfully leverages semantic sparsity to reduce redundancy in reasoning trajectories.
>
> **Table JM3v-3: Embedding-based redundancy rate (ratios of sentences with more than N similar sentences during generation) reduction with RPC**
>
> | |       N=1       |    |       N=2       | |       N=4       |   |
> |-----------------------|-----------------|---------|-----------------|---------|-----------------|---------|
> |  | Full KV | RPC     | Full KV  | RPC  | Full KV  | RPC     |
> | R1-Distill-Qwen-7B | 65.5%   | 33.8%   | 48.9%   | 15.4%   | 29.8%    | 4.1%    |
> | R1-Distill-Llama-8B  | 69.7%     | 33.8%   | 52.7%   | 15.4%   | 35.0%  | 4.1%    |
> | QwQ-32B   | 66.0%   | 28.2%   | 48.0%| 13.3%   | 29.6% | 5.3%    |
>
> In the revised version of our paper, we will include both comprehensive token selection visualizations and the full embedding-based redundancy analysis in the appendix. We believe this will help readers more clearly appreciate the effectiveness of RPC in eliminating redundancy while preserving critical reasoning information.
>
> Once again, we sincerely thank the reviewer for this helpful suggestion.
>
> [1] Jiaming Tang, , Yilong Zhao, Kan Zhu, Guangxuan Xiao, Baris Kasikci, Song Han. "QUEST: Query-Aware Sparsity for Efficient Long-Context LLM Inference." ICML. 2024.
>
> [2] Oren, Matanel, et al. "Transformers are multi-state rnns." arXiv preprint arXiv:2401.06104 (2024).
>
> [3] Zhang, Zhenyu, et al. "H2o: Heavy-hitter oracle for efficient generative inference of large language models." Advances in Neural Information Processing Systems 36 (2023): 34661-34710.

---

> > ### Comment · Reviewer_JM3v · 2025-08-04
> >
> > Thank you for your response. Your explanations were clear and successfully addressed my previous questions. Accordingly, I have raised my score by one point.

---

> ### Author Response · Authors · 2025-08-05
>
> Thank you very much for your positive response and raising the rating score.

---

### Official Review · Reviewer_r668 · 2025-06-28

**Clarity:** 3
**Significance:** 3
**Originality:** 3
**Rating:** 5
**Confidence:** 2

**Summary:**

This paper proposes Reasoning Path Compression (RPC), a training-free KV cache compression method aimed at improving the inference efficiency of reasoning-focused LLMs that generate long intermediate reasoning paths. The authors observe that such reasoning paths often exhibit semantic sparsity (e.g., redundant or repeated logical steps) and leverage this property by periodically compressing the KV cache, retaining only tokens that receive high importance scores computed using recent queries as a selector window. The method is evaluated on the DeepSeek-R1-Distill-Qwen-7B and QwQ-32B models across benchmarks including AIME 2024, LiveCodeBench, and IFEval, and is compared against prior methods such as LightThinker, H2O, and TOVA.

**Questions:**

### Questions and Suggestions for the Authors


1. **Further empirical investigation of semantic sparsity**
   The motivation around semantic sparsity currently relies on n-gram Shannon entropy measurements using a single model. Have the authors conducted further empirical investigations of semantic sparsity on other reasoning LLMs and with other metrics, such as embedding-based similarity between reasoning steps, to measure redundancy beyond exact n-gram repetition? Additionally, do similar redundancy patterns appear on coding benchmarks such as LiveCodeBench?

2. **Ablation and justification for the selector window**
   The paper highlights the selector window as a key component, asserting that attention scores relative to recent tokens effectively indicate the relevance of previous tokens. While an ablation on different selector window sizes is provided, it would be helpful to clarify how this empirically supports the claim that using attention scores over recent tokens is an effective method for importance estimation, versus alternatives like average attention across all tokens.

3. **Clarification on comparison with SnapKV and HeadKV**
   The paper does not include SnapKV and HeadKV in the experimental comparisons. Since LLM generation is autoregressive and both the prompt and generated tokens contribute to the autoregressive input for the next token, could the authors clarify why these methods are not included and how the differences between compressing prompts and compressing generated reasoning paths practically manifest in the context of these methods?

4. **Clarification on the accuracy drop with larger R**
   The paper mentions that using a larger selector window size (R) can lead to an accuracy decrease, but the explanation provided is not fully clear. Could the authors elaborate on why evaluating importance over a larger set of recent tokens would decrease performance?

---

**Criteria for potential score increase**:
A response that includes a deeper investgation of semantic sparsity across models and metrics, and clarifications on the observed behaviors (such as the accuracy drop with larger R) would strengthen the paper’s motivation and technical justification. This would positively impact my evaluation of the paper’s quality and clarity.

**Ethical Concerns:**

["NO or VERY MINOR ethics concerns only"]

**Final Justification:**

My main concern was the simplicity of the sparsity analysis in the paper, which used simple n-gram Shannon entropy to measure redundancy, which is appropriate for detecting exact or near-exact repetition, but may miss paraphrased or logically equivalent content.

Given that the authors have conducted some more appropriate sparsity analysis using semantics in the rebuttal, I now recommend accept. However, I would strongly suggest that the authors include the new semantic sparsity analysis in their manuscript.

**Limitations:**

The paper could benefit from noting that RPC is primarily tested on math and coding tasks using DeepSeek and Qwen models, and its generalization to other tasks is unclear. Additionally, while RPC is training-free, tuning hyperparameters (e.g., compression interval, selector window) may require task- or model-specific calibration to maintain accuracy.

**Quality:**

3

**Strengths And Weaknesses:**

### Strengths

- The motivation of the paper is clear: reasoning chains in LLMs are long and often contain semantically redundant steps, which increases inference cost.
- The proposed RPC method is training-free and shows good empirical performance gains, improving throughput and reducing memory usage with only modest accuracy drops.
- The paper evaluates the method on well-chosen benchmrks (AIME 2024, LiveCodeBench, IFEval) using two models (DeepSeek-R1-Distill-Qwen-7B and QwQ-32B), which supports the validity of the findings.
- The ablation studies are informative, illustrating how compression interval and selector window size affect the accuracy-efficiency trade-off, which is useful for practitioners.

### Weaknesses

- The claim that prior KV compression methods are primarily designed for long input/short output settings (e.g., line 93) would benefit from additional justification or empirical validation to clarify the limits of prior methods.
- While the empirical results are strong, the motivation regarding semantic sparsity could be investigated more deeply. The current analysis uses n-gram Shannon entropy to measure redundancy, which is appropriate for detecting exact or near-exact repetition, but may miss paraphrased or logically equivalent content. Incorporating embedding-based similarity measures between reasoning steps, commonly used for semantic redundancy detection, would strengthen the analysis.
- The redundancy analysis appears to primarily use DeepSeek-R1-Distill-Llama-8B outputs. Given that outputs from reasoning models can vary significantly, especially on reasoning-heavy tasks, additional analysis using a broader set of models would strengthen the motivation and generality of the semantic sparsity observation.
- A stronger empirical investigation of semantic sparsity would help the reader understand the problem space more thoroughly and clarify why the proposed method works as well as it does.
- Minor clarity issue: there is a typo on line 264 (“comperssing”).

---

> ### Author Rebuttal · Authors · 2025-07-31
>
> **[Response r668-1] Weakness 1 and Question 3: Justification for the unsuitability of existing methods in long generation scenario**
>
> As discussed in **Section 2.2** of our manuscript, existing KV compression techniques can be categorized into two types:
> 1. those that compress only the input prompt
> 2. those that also compress the generated tokens
>
> We now explain why both categories fail to address the challenges posed by reasoning models.
>
> **1. Methods that compress only the input prompt (e.g., SnapKV, HeadKV):**
> These methods are designed to compress long input prompts only, so they do not apply any compression to the KV cache associated with newly generated tokens.
> However, reasoning tasks feature relatively short inputs but very long outputs. As shown in **Table r668-1**, the average input length is approximately 115 tokens, while the average output length exceeds 13K tokens. In this setting, compressing the input tokens brings negligible benefit, as they make up only a small fraction of the total KV cache. (Further discussion on the difference between compressing input/output tokens: **Response r688-6**)
>
> **Table r668-1 Input/output length statistics of AIME 2024 with QwQ-32B**
> | | Mean | Std  |
> |-----------|----------|--------|
> | Input     | 115.5    | 38.1   |
> | Output    | 13834.6  | 7365.5 |
>
> **2. Methods that compress generated tokens (e.g., H2O, TOVA):**
> To address generative compression, H2O and TOVA maintain a fixed-size cache during generation and evict tokens based on attention scores. While effective for standard language modeling tasks with predictable and relatively short outputs, they struggle with reasoning tasks for the following reasons:
> - Output lengths in reasoning tasks are not only much longer but also highly variable across samples. As a result, choosing an appropriate cache budget becomes difficult.
> - These methods also evict tokens one-by-one at each decoding step. This incurs substantial computational overhead and may disrupt long-term semantic coherence.
>
> On the other hand, the proposed RPC addresses these limitations in two key ways:
>  RPC defines a compression ratio rather than a fixed KV cache size, enabling adaptive budget allocation based on the actual number of generated tokens.
> Second, RPC performs periodic compression at fixed intervals, reducing computational overhead while preserving semantic consistency across long reasoning paths.
>
> We provide accuracy comparison in **Table 1** of our manuscript. To further support our claim, we extended the evaluation to additional benchmarks and models in **Table r668-2**. Here, previous works consistently demonstrate degraded performance, while RPC remains robust.
>
> **Table r668-2 Accuracy evaluation with R1-Distill-Qwen-7B**
> |   | AIME 2024 | LiveCodeBench | IFEval |
> |------------------|-----------|---------------|--------|
> | Full KV   | 55.5  | 37.6  | 55.1   |
> | H2O (avg)    | 42.5   | 22.5    | 51.8   |
> | TOVA (avg)  | 42.5   | 21.5    | 48.8   |
> | LightThinker | 6.7 | 0.7  | 25.1   |
> | RPC (P=4096) | 52.9    | 35.9   | 56.6   |
> | RPC (P=1024) | 50.4    | 33.5  | 57.3   |
>
> **[Response r668-2] Weakness 2/3/4 and Question 1: Additional investigation on semantic sparsity**
>
> **Comprehensive Analysis of Semantic Sparsity**
>
> We conducted a deeper investigation of semantic sparsity using embedding-based similarity analysis. Specifically, we applied sentence-level vector embeddings to the generated outputs and computed pairwise cosine similarity between sentence embeddings. Two sentences were considered “semantically similar” if their cosine similarity exceeded 0.75. Based on this, we defined the **redundancy rate** as the proportion of sentences that have more than **N(N=1,2,4) semantically similar sentences** within the same output.
>
> In addition, as you recommended, we expanded the analysis to a broader set of models (three reasoning LLMs and two non-reasoning LLMs). The analysis of reasoning and non-reasoning LLMs used AIME 2024 and  HelloBench dataset, respectively.
>
> As shown in **Table r668-3**, reasoning LLMs exhibit higher redundancy rates compared to non-reasoning LLMs. This indicates that reasoning LLMs generate more semantically similar output than non-reasoning LLMs.
>
> **Table r668-3 Embedding-based redundancy rate comparison between reasoning and non-reasoning LLMs**
> |   |     | N=1   | N=2   | N=4   |
> |---------------|---------------|-------|-------|-------|
> | Reasoning     | R1-Distill-Qwen-7B  | 65.5% | 48.9% | 29.8% |
> |  | R1-Distill-Llama-8B | 69.7% | 52.7% | 35.0% |
> |   | QwQ-32B     | 66.0% | 48.0% | 29.6% |
> | Non-Reasoning | LongWriter-8B      | 44.5% | 32.5% | 22.2% |
> |   | LongWriter-9B     | 33.8% | 19.8% | 12.0% |
>
> **Impact of RPC in reducing semantic redundancy**
>
> We examined whether semantic redundancy is effectively reduced by RPC.
>
> As shown in **Table r668-4**, our analysis reveals that the redundancy rate dropped significantly after applying RPC. This demonstrates that RPC successfully reduces redundancy in reasoning trajectories.
> We sincerely appreciate your suggestion to include a more quantitative analysis of semantic sparsity. These new results and discussions will be updated in the revised version of the manuscript.
>
> **Table r668-4 Embedding-based redundancy rate reduction with RPC**
> |  |   N=1  |    |   N=2  |  |  N=4   |    |
> |-----------------------|---------------|---------|---------------|---------|---------------|---------|
> |  |Full KV| RPC| Full KV| RPC|Full KV |RPC|
> | R1-Distill-Qwen-7B| 65.5%| 33.8%| 48.9%| 15.4%| 29.8%| 4.1%|
> | R1-Distill-Llama-8B| 69.7%| 33.8%| 52.7%| 15.4%| 35.0%| 4.1%|
> | QwQ-32B| 66.0%| 28.2%| 48.0%| 13.3%| 29.6%| 5.3%|
>
> We will also update these new results and discussions in our manuscript.
>
>
> **[Response  r668-3] Weakness 5: Typo**
>
> Thank you for catching this typo. We will correct it.
>
> **[Response r668-4] Question 1: Redundancy pattern on coding benchmark**
>
> We observe that redundancy patterns similar to those found in mathematical reasoning tasks also frequently appear in code generation. When reasoning LLMs are applied to coding problems, the generated outputs often include repeated explanations of core algorithmic concepts, multiple reconfirmations of solution steps, and restatements of reasoning strategies.
>
> To quantify this redundancy, we applied the same embedding-based similarity analysis described in **Response r668-2**. As shown in **Table r668-5**, the redundancy level observed in LiveCodeBench is even higher than that measured in AIME 2024. This observation reinforces our claim that semantic sparsity is a general property of reasoning LLM outputs.
>
> **Table r668-5 Embedding-based redundancy rate of R1-Distill-Qwen-7B**
>
> | | N=1   | N=2   | N=4   |
> |---------------|-------|-------|-------|
> | AIME 2024| 65.5% | 48.9% | 29.8% |
> | LiveCodeBench | 70.9% | 54.3% | 35.8% |
>
> **[Response r668-5] Question 2: Justification for the selector window**
>
> As shown in **Table 3** of our manuscript, using a large selector window size of R=128 results in lower accuracy compared to R=32. This result indicates that when the selector window includes outdated tokens, these tokens may interfere identifying truly relevant tokens for the current reasoning step. Consequently, we can expect that using attention from all previously generated tokens is likely to degrade accuracy. More importantly, accumulating attention scores from all tokens is not a practical solution in reasoning tasks with long output sequences, as it introduces substantial computational overhead.
>
> To explore alternative strategies, we experimented with using a selector window composed of the last few tokens of the input prompt. This alternative led to a slight decrease in accuracy (see **Table r668-6**). While using the prompt's final tokens as selectors can be advantageous in capturing prompt-relevant context, our results suggest that leveraging recently generated tokens—those most pertinent to the current reasoning step—yields more robust and accurate token selection for KV retention.
>
> **Table R668-6 Accuracy of R1-Distill-Qwen-7B after applying RPC with recent or prompt tokens as selector window**
> | | P=4096 | P=1024|
> |------------------|---------------------------|----------------------------|
> |Recent Tokens|52.9|50.4|
> |Prompt Tokens|42.5|45.0|
>
>
> **[Response r668-6] Question 3: Difference of compressing input tokens and output tokens**
>
> For input tokens, the entire sequence is available from the beginning of inference and remains fixed, making it possible to compress the input context in a single step without dynamic updates.
>
> In contrast, output tokens are generated autoregressively and their final length is unknown in advance. The importance of each generated token evolves as reasoning progresses, so token selection must be performed dynamically during generation.
>
> It is important to note that methods such as SnapKV and HeadKV are designed specifically for compressing input tokens and do not provide mechanisms for compressing dynamically generated output tokens. Since our focus is on efficient long-form reasoning generation, where output token compression is essential, we did not include SnapKV and HeadKV in our experimental comparisons.
>
> **[Response r668-7] Question 4: Clarification on the accuracy drop with larger R**
>
> As mentioned earlier, the motivation for using recently generated tokens as the selector window is to ensure that token retention decisions reflect the current stage of reasoning. By focusing on attention from the most recent tokens, RPC can better identify which past tokens remain relevant.
>
> However, if the selector window size R becomes too large (e.g., 128), it starts to include many older tokens whose attention patterns may no longer match the current reasoning context. Involving these outdated tokens in importance estimation can distort the selection process, leading to the retention of non-critical tokens and possibly discarding those essential for the present reasoning step.

---

> > ### Comment · Reviewer_r668 · 2025-08-05
> >
> > I am satisfied with the authors' response and appreciate their additional analysis. I will recommend acceptance provided that the authors include the new semantic sparsity analysis in their manuscript.

---

> > > ### Author Response · Authors · 2025-08-05
> > >
> > > Thank you for your positive feedback. We will make sure to include the new semantic sparsity analysis in our manuscript.

---

### Official Review · Reviewer_jmzr · 2025-07-03

**Clarity:** 3
**Significance:** 2
**Originality:** 2
**Rating:** 4
**Confidence:** 2

**Summary:**

Based on the redundant segments and inherent semantic sparsity in reasoning LLMs, this paper introduces a reasoning  compression method RPC.  RPC leverages this characteristic by periodically compressing the KV cache and employs an importance scoring mechanism based on a selector window composed of recent queries. Experimental results validate the effectiveness of the proposed method, e.g., demonstrating that RPC compress KV cache 4$\times$ with accuracy degradation limited to 1.2\%.

**Questions:**

Please see more details in Weaknesses. BTW, the proposed algorithm involves many hyperparameters. So, could these hyperparameters be learned from an optimization objective?

**Ethical Concerns:**

["NO or VERY MINOR ethics concerns only"]

**Final Justification:**

I'd like to raise the score from 3 to 4.

**Limitations:**

yes

**Paper Formatting Concerns:**

There is no major formatting issues.

**Quality:**

3

**Strengths And Weaknesses:**

Strengths:
1. This paper is well-written and easy to follow.
2. The proposed method has engineering impact. For example,  KV‐cache is significantly reduced without fine-tuning model weights.
3. The experimental evaluations are relatively sufficient, although more baseline methods are encouraged to be involved.

Weaknesses:
1. The proposed algorithm involves many hyperparameters, and in practical applications you need to perform a fine-grained grid search to achieve good results. So, this method seems not flexible enough.
2. In practice, different heads produce different attention scores. Some heads play a critical role at specific reasoning stages, but from this method, their peaks can be drowned out by moderate scores from other heads, causing them to fall out of the Top-P after aggregation. This  can potentially lead to loss of fine-grained information.
3. Does it make sense to rely on attention weights as the core metric?
4. I feel the paper includes too few baselines, considering only the full CV compression method. Could more baseline methods be evaluated? (I’m not a KV-cache specialist myself. Sorry I can’t suggest specific references.)

---

> ### Author Rebuttal · Authors · 2025-07-31
>
> **[Response jmzr-1] Weakness 1 and Question 1: Selecting hyperparameters**
>
> Thank you for your valuable comment. As you pointed out, the proposed RPC method involves setting two hyperparameters, compression interval (P) and selector window size (R), to minimize accuracy degradation after compression. In our experiments, we used grid search to identify appropriate values.
>
> However, we would like to clarify that **the sensitivity of these hyperparameters is relatively low, and fine-grained tuning is not necessary**. This is why our grid search was conducted over a coarse set of values, as shown in Figure 7 and Table 3. For example, in Table 3, we evaluated R using only four values (1, 8, 32, 128), and the best accuracy was achieved at R=32. Slight variations (e.g., R=40) do not significantly impact accuracy, so we limited our evaluation to powers of two.
>
> Moreover, **both P and R exhibit consistent and predictable trends**, which further simplifies the tuning process. Since P controls how frequently compression occurs, it is expected to saturate once enough tokens are accumulated for effective compression. Similarly, R determines the size of the context window for importance estimation: too small an R may miss relevant information, while too large an R may include outdated context. As a result, accuracy tends to improve initially with increasing R, reaches a peak, and then degrades as R becomes excessively large. These intuitive trends allow users to tune P and R without needing an exhaustive search.
> Importantly, the hyperparameter setup exhibits good cross-task transferability, as the values selected for AIME2024 generalize well to other benchmarks. (In our experiments, we used P=1024 or 4096 and R=32 across all benchmarks.) While some tuning may be required for tasks with significantly different output length distributions, the process remains straightforward and does not involve burdensome fine-grained search.
>
> Because the effects of hyperparameters (P and R) are easy to interpret and manage, we believe that additional learning-based optimization is unnecessary. However, we see potential to eliminate the need for the compression interval P. Specifically, RPC could be further improved by incorporating a mechanism that dynamically detects when a sufficient number of tokens have been generated to trigger compression, rather than relying on a fixed interval. We consider this a promising direction for future work.
> In summary, only P and R function as true hyperparameters in RPC, and both can be set in a simple and intuitive manner for effective and robust performance.
>
> **[Response jmzr-2] Weakness 2: Granularity of token selection and attention score aggregation**
>
> As you pointed out, different attention heads often play different roles. However, from the perspective of selecting important tokens for KV cache compression, a **coarser-grained token selection strategy**, such as **layer-wise selection** as adopted in RPC, offers clear advantages in preserving accuracy.
>
> To support this, **Table jmzr-1** compares accuracy after KV cache compression using different granularities of token selection: **layer-wise**, **key-value-group-wise**, and **attention head-wise**. The results show that layer-wise token selection, which aggregates attention scores across all heads within a layer, consistently yields the highest accuracy.
>
> **Table jmzr-1 AIME 2024 pass@1 comparison between attention score aggregation granularities**
> | AIME 2024 (pass@1)      | Layer Aggregation | Group Aggregation | Head (No Aggregation) |
> |-------------------------|-------------------|-------------------|----------------------|
> | R1-Distill-Qwen-7B (P=4096)          | 52.9              | 50.8              | 49.6                 |
> | R1-Distill-Qwen-7B (P=1024)          | 50.4              | 50.4              | 47.5                 |
> | QwQ-32B (P=4096)        | 78.3              | 75.0              | -                    |
> | QwQ-32B (P=1024)        | 78.3              | 77.5              | -                    |
>
> These findings indicate that finer-grained selection at the head level does not necessarily lead to better accuracy in KV compression. One possible reason is that **attention scores from a single head may not contain sufficient semantic information** to identify truly important tokens. A token strongly attended by one head may not correspond to meaningful context and could even reflect noise. In contrast, aggregating attention scores across all heads in a layer allows for a more semantically informed estimate of token importance.
>
> This observation, advantage of coarse-grained token selection, is also supported by previous work. **AdaKV** [1] updated its method to use group-level selection rather than head-level selection. Their experiments showed that group-level and head-level selection yielded nearly identical accuracy on **LongBench**. Additionally, **TOVA** [2] reported that for both TOVA and **H2O** [3], **layer-level token selection** outperformed head-level selection in terms of perplexity.
>
> Moreover, in models using **grouped-query attention (GQA)**, multiple attention heads share a single KV head. Performing token selection separately for each attention head in such models would require maintaining separate KV caches per head. This introduces additional memory overhead, potentially defeating the purpose of compression. (Due to this overhead, it was not feasible to fully evaluate the accuracy of **QwQ** under head-wise token selection, and thus the corresponding entry in **Table jmzr-1** is left blank.) Therefore, **even aside from accuracy**, head-level selection is not a practical option for GQA-based models.
>
> Overall, the **layer-level token selection strategy** used in RPC is a practical approach for compressing the KV cache of reasoning LLMs.
>
>
> **[Response jmzr-3] Weakness 3: Why we use attention weights as our core metric**
>
> Because attention weights are directly used to merge token information (specifically, the value vectors in the KV pairs), they reflect the relative influence of each token on subsequent outputs. Therefore, attention weights can serve as a reasonable proxy for token importance.
>
> Building on this intuition, many prior KV cache compression techniques (e.g., SnapKV [4], H2O[3]) have also leveraged attention weights to identify and retain important tokens.
> In summary, leveraging attention weights for token screening is not only intuitive but also a well-established practice in KV compression research, and RPC extends this paradigm to the unique characteristics of reasoning LLMs.
>
> **[Response to jmzr-4] Weakness 4: Additional evaluation results of baselines**
>
> As you noted, while **Table 1** in the main paper compares RPC with the full KV cache as well as baseline methods (H2O, TOVA, and LightThinker), Table 2 only compares RPC with full KV Cache.
>
> We agree that a more comprehensive comparison is more desirable. To address this, we have evaluated the baselines across all three benchmarks (AIME 2024, LiveCodeBench, and IFEval) using **R1-Distill-Qwen-7B**, and the results are now provided in **Table jmzr-2**.
>
> **Table jmzr-2 Additional evaluation results of baselines and comparison with RPC (R1-Distill-Qwen-7B)**
> |                  | AIME 2024 | LiveCodeBench | IFEval |
> |------------------|-----------|---------------|--------|
> | Full KV          | 55.5      | 37.6          | 55.1   |
> | H2O (avg)        | 42.5      | 22.5          | 51.8   |
> | TOVA (avg)       | 42.5      | 21.5          | 48.8   |
> | LightThinker     | 6.7       | 0.7           | 25.1   |
> | RPC (P=4096)     | 52.9      | 35.9          | 56.6   |
> | RPC (P=1024)     | 50.4      | 33.5          | 57.3   |
>
> For this extended evaluation, we allocated 25% of the average output length (measured from full KV cache setup) as a fixed cache size for H2O(avg) and TOVA(avg). Please note that this approach produces slightly different results (yet with same trend) from the per-sample cache budgeting strategy used in our origianal manuscript, which would require first measuring the generation lengths of the reasoning LLMs under full KV cache settings and then determining the KV cache budget as 25% of each individual length.
> Running H2O and TOVA with such a strategy is extremely slow due to their inefficiencies and token-by-token compression mechanisms, so it was impractical to adopt the per-sample cache budgeting strategy for this extended evaluation.
>
> Even under this more comprehensive evaluation, RPC maintains its advantage by consistently preserving accuracy better than previous KV compression techniques.
>
> We will also update **Table 2** in the revised version of the paper to reflect the new results.
>
> [1] Feng, Yuan, et al. "Ada-kv: Optimizing kv cache eviction by adaptive budget allocation for efficient llm inference." arXiv preprint arXiv:2407.11550 (2024).
>
> [2] Oren, Matanel, et al. "Transformers are multi-state rnns." arXiv preprint arXiv:2401.06104 (2024).
>
> [3] Zhang, Zhenyu, et al. "H2o: Heavy-hitter oracle for efficient generative inference of large language models." Advances in Neural Information Processing Systems 36 (2023): 34661-34710.
>
> [4] Li, Yuhong, et al. "Snapkv: Llm knows what you are looking for before generation." Advances in Neural Information Processing Systems 37 (2024): 22947-22970.

---

> > ### Comment · Reviewer_jmzr · 2025-08-05
> >
> > Thanks for your effort and detailed response. However, I keep my original score because of several reasons
> > 1. The rebuttal relies on evaluations over a narrow set of values (e.g., $R \in \{1, 8, 32, 128\}, P \in \{1024, 4096\}$), which is insufficient to claim low sensitivity. Additionally, in Table 3, accuracy fluctuations of more than 5% are observed when varying R, indicating non-trivial sensitivity.
> > 2.  Is this method flexible to incorporate to other metrics? If so,  the authors could have experimented with a hybrid scoring mechanism combining attention and other metrics like uncertainty-based signals, which may offer more stable compression.

---

> ### Author Response · Authors · 2025-08-07
>
> While we acknowledge the reviewer’s position, we respectfully submit one additional clarification in the interest of ensuring the technical discussion is complete.
>
>
> **[Response to Additional Question 1]**
>
> **1) Regarding accuracy fluctuations across R:**
>
> While **Table 3** shows accuracy differences of several percentage points as R varies, we would like to emphasize that RPC, even under strong compression (4×), consistently outperforms all baseline methods in **Table 1**—including H2O, TOVA, and LightThinker in the worst case across the tested R values. The stability and competitiveness of RPC, despite such fluctuations, demonstrate that our approach provides a robust improvement over existing baselines.
>
> **2) Regarding the range of hyperparameter sweeps:**
>
>  Our ablation studies for P were conducted over a wide range of values (from 4 up to 16,384), as shown in **Figure 7**, not just the two values used in the main experiments. This extensive sweep allowed us to clearly illustrate the trends in accuracy, throughput, and memory usage, and helped us select P=1024 and 4096 as practical trade-offs for accuracy and efficiency. Similarly, for R, we searched across a broad range that spans from single-token granularity up to values that cover several words, full sentences, and even beyond. This comprehensive sweep ensured that our conclusions about practical and robust settings for P and R are well supported by empirical evidence.
>
> **3) Interpreting the impact of P and R:**
>
> We observe that the largest accuracy degradation occurs when P or R is set to extremely small values. This is intuitive, as overly frequent compression (very small P) or an excessively narrow context for importance estimation (very small R) causes the algorithm to depend too heavily on a small set of tokens, leading to unstable retention of critical information. In practice, simply avoiding these extreme cases is sufficient to prevent significant accuracy loss—accuracy stabilizes quickly as P and R are increased to modest values.
>
> On the other hand, excessively large values of P or R naturally lead to diminished efficiency due to increased memory overhead and lower throughput. Therefore, in practical settings, P and R can be easily chosen from a reasonable range that avoids both extremes, enabling robust accuracy and efficiency trade-offs without the need for extensive hyperparameter tuning.
>
> **[Response to Additional Question 2]**
>
> As you pointed out, RPC can be extended to incorporate additional metrics beyond attention-based scoring. Implementing another scoring approach is straightforward within our framework, as the importance scoring step is modular.
>
> In this work, we chose to focus on attention-based scoring because it provides a direct, interpretable measure of each token’s relevance to ongoing reasoning. Our experiments presented in this paper demonstrate that this mechanism alone is sufficient to effectively identify and remove redundant KV entries,  leading to substantial improvements in memory and throughput with minimal impact on accuracy.
>
> In particular, we empirically validated the effectiveness of our attention-based scoring mechanism for redundancy reduction. We applied sentence-level vector embeddings to the generated outputs and computed pairwise cosine similarity between sentence embeddings, defining sentences as **semantically similar** if their cosine similarity exceeded 0.75. The redundancy rate was then calculated as the proportion of sentences with more than N (N=1,2,4) semantically similar sentences in the same output.
>
> As summarized in **Table jmzr-3**, this analysis—conducted on outputs from three reasoning LLMs (R1-Distill-Qwen-7B, R1-Distill-Llama-8B, and QwQ-32B) using the AIME 2024 dataset—shows that the redundancy rate dropped significantly after applying RPC.
>
> **Table jmzr-3 Embedding-based redundancy rate (ratios of sentences with more than N similar sentences during generation) reduction with RPC**
> |                  |   N=1   |        |   N=2   |        |   N=4   |        |
> |------------------|---------|--------|---------|--------|---------|--------|
> |                  | Full KV |  RPC   | Full KV |  RPC   | Full KV |  RPC   |
> | R1-Distill-Qwen-7B      | 65.5%   | 33.8%  | 48.9%   | 15.4%  | 29.8%   | 4.1%   |
> | R1-Distill-Llama-8B     | 69.7%   | 33.8%  | 52.7%   | 15.4%  | 35.0%   | 4.1%   |
> | QwQ-32B                | 66.0%   | 28.2%  | 48.0%   | 13.3%  | 29.6%   | 5.3%   |
>
> This provides strong empirical evidence that the current attention-based scoring method effectively removes redundant content by leveraging semantic sparsity in the reasoning process.
>
> We appreciate your careful review and constructive comments.

---

> > ### Comment · Reviewer_jmzr · 2025-08-08
> >
> > Thanks for your detailed response, which addressed my concerns. I'd like to raise the score from 3 to 4.

---

> ### Author Response · Authors · 2025-08-09
>
> Thank you for your thoughtful follow-up and increasing the score.

---

### Official Review · Reviewer_SCoz · 2025-07-03

**Clarity:** 3
**Significance:** 3
**Originality:** 2
**Rating:** 5
**Confidence:** 3

**Summary:**

This paper introduces Reasoning Path Compression (RPC), a training-free method to accelerate inference in reasoning-focused language models that generate long intermediate reasoning paths. The key insight is that reasoning sequences exhibit "semantic sparsity" - they often contain redundant steps, repeated logic, and low information density relative to their length. RPC periodically compresses the KV cache by retaining only high-importance tokens (determined by attention scores), achieving up to 1.60× throughput improvement on QwQ-32B with only 1.2% accuracy drop, making reasoning LLMs more practical for deployment.

**Questions:**

What is the distribution of the response length in the reasoning model? How often will the compression be triggered under the large interval P=4096?

**Ethical Concerns:**

["NO or VERY MINOR ethics concerns only"]

**Final Justification:**

Thank the authors for the detailed responses and additional experiments. They have solved my concerns, and I will raise my score accordingly.

**Limitations:**

Authors have discussed their limitations and social impact in the paper

**Quality:**

3

**Strengths And Weaknesses:**

Strengths:

* This paper first investigates the semantic sparsity in the reasoning models: the long chain of thoughts often contains redundant messages with little information. It defines the semantic sparsity as the n-gram Shannon entropy and finds that the entropy is much lower in the reasoning models than the generic purpose LLMs in various response lengths. This motivates the compression of the response in reasoning models.
* Compared with other baselines, RPC can keep comparable performance with a 4.0x compression ratio on the KV Cache in DeepSeek-R1-Distill-Qwen-7B and QwQ-32B. It results in a higher throughput and lower peak memory during the inference.
* It provides detailed analyses on the important hyperparameters such as the selector window size R and the compression interval P to investigate their impact on the throughput and peak memory.

Weaknesses:
* From Figure 7, we can find that the accuracy significantly drops when the compression interval P is small. To achieve comparable results, it requires a large P such as 1024. Therefore, responses that are shorter than 2048 tokens can only trigger at most one compression when P=1024. It is important to demonstrate the distribution of the DeepSeek-R1-Distill-Qwen-7B's and QwQ-32B's response length in the AIME 2024, LiveCodeBench, and IFEval datasets. If the response lengths are significantly smaller than P=4096 or 1024, the comparable accuracy in Table 2 may be because most of the test cases do not trigger compression at all, rather than good performance of RPC.
* As discussed in Figures 1 and 3, there is some redundant information in the long reasoning steps, and the goal of the compression is to remove the redundancy. So it is important to demonstrate what kind of tokens have been compressed based on the proposed Important Token Selection. The newly generated tokens may have a higher attention score than the previous similar tokens.

Others:
Figure 6 is a little vague when zoomed in

---

> ### Author Rebuttal · Authors · 2025-07-31
>
> **[Response SCoz-1]  Weakness 1 and Question 1:  Output length distribution of reasoning LLMs**
>
> Thank you for raising the important point about the relationship between output length and the frequency of compression triggers in RPC.
>
> In **Tables SCoz-1 and SCoz-2**, we provide detailed statistics on the output lengths of R1-Distill-Qwen-7B and QwQ-32B for all three benchmarks used in our evaluation (AIME 2024, LiveCodeBench, and IFEval).
>
> For AIME 2024 and LiveCodeBench, the mean output length ranges from 12K to 14K tokens, with a maximum of 32K tokens. In such cases, the proposed RPC with P=1024 or 4096 effectively mitigates rapid KV cache growth.
>
> For IFEval, while the maximum still reaches 32K tokens, the mean output length is below 2K tokens. As you correctly pointed out, compression is rarely triggered when P=1024 or 4096 in this setting.
>
> However, as shown in **Table SCoz-3**, because the average output length of IFEval is much shorter than that of AIME 2024 and LiveCodeBench, using a smaller P value (such as 256 or 512) can still preserve accuracy effectively. Nonetheless, we chose to use a consistent P value of 1024 or 4096 across all benchmarks, as our primary objective is to prevent excessive KV cache expansion caused by unexpectedly long reasoning paths.
>
>
> **Table SCoz-1 Output length distribution (R1-Distill-Qwen-7B)**
>
>
> | R1-Distil-Qwen-7B | Mean     | Min   | Max    | Std     |
> |----------------------------|----------|-------|--------|---------|
> | AIME 2024                  | 13668.6  | 2413  | 32768  | 9356.4  |
> | LiveCodeBench              | 11889.1  | 809   | 32768  | 7066.2  |
> | IFEval                     | 1778.4   | 190   | 32768  | 5073.3  |
>
> **Table SCoz-2 Output length distribution (QwQ-32B)**
> | QwQ-32B         | Mean     | Min   | Max    | Std     |
> |-----------------|----------|-------|--------|---------|
> | AIME 2024       | 13834.6  | 2747  | 32768  | 7365.5  |
> | LiveCodeBench   | 13454.6  | 491   | 32768  | 9692.1  |
> | IFEval          | 1336.9   | 144   | 32768  | 1844.3  |
>
> **Table SCoz-3 IFEval pass@1**
> | Model                        | Full KV | P=256 | P=512 | P=1024 | P=4096 |
> |------------------------------|---------|-------|-------|--------|--------|
> | R1-Distill-Qwen-7B  | 55.1    | 57.1  | 56.7  | 57.3   | 56.6   |
>
>
> **[Response SCoz-2] Weakness 2: Analysis on evicted tokens with an example**
>
> Thank you for your valuable comment. To evaluate whether the proposed RPC effectively removes redundancy from long reasoning sequences, we provide both a concrete example of token selection and quantitative evidence supporting the removal of redundant tokens.
>
> First, to illustrate how RPC selects important tokens, we visualized the tokens retained by RPC using **QwQ-32B** on a sample from the **AIME 2024** dataset. Since the set of selected tokens can vary across layers in RPC, we highlighted in bold the tokens that were selected by more than 50% of the layers in **Example SCoz-1 (Shown at the bottom of this rebuttal)**. (Given the long sequence lengths produced by reasoning models, it presents a representative subset of the output.)
>
> From this example, we observe that the tokens most frequently retained by RPC fall into two primary categories:
>
> - **Tokens containing critical information that advances the reasoning process**, such as conclusions of intermediate steps.
>
>
> - **Tokens indicating transitions between reasoning steps**, which, although not semantically rich on their own, play an important contextual role when processed through the attention mechanism, as they help align and integrate information across reasoning stages.
>
>
> This qualitative analysis demonstrates that RPC effectively preserves essential reasoning content while removing redundant or outdated derivations.
>
> Second, to further validate RPC's ability to reduce redundancy, we performed an additional quantitative analysis using embedding-based sentence similarity.
>
> Specifically, we applied sentence-level vector embeddings to the generated outputs and computed pairwise cosine similarity between sentence embeddings. Two sentences were considered “semantically similar” if their cosine similarity exceeded 0.75. Based on this, we defined the redundancy rate as the proportion of sentences that have more than N(N=1,2,4) semantically similar sentences within the same output.
> This analysis was conducted on outputs from three reasoning LLMs (R1-Distill-Qwen-7B, R1-Distill-Llama-8B, and QwQ-32B) using the AIME 2024 dataset.
>
> As shown in **Table SCoz-4**, our analysis reveals that the redundancy rate dropped significantly after applying RPC. This result provides strong evidence that RPC successfully leverages **semantic sparsity** to reduce redundancy in reasoning trajectories.
>
> **Table SCoz-4 Embedding-based redundancy rate (ratios of sentences with more than N similar sentences during generation) reduction with RPC**
> |                  |   N=1   |        |   N=2   |        |   N=4   |        |
> |------------------|---------|--------|---------|--------|---------|--------|
> |                  | Full KV |  RPC   | Full KV |  RPC   | Full KV |  RPC   |
> | R1-Distill-Qwen-7B      | 65.5%   | 33.8%  | 48.9%   | 15.4%  | 29.8%   | 4.1%   |
> | R1-Distill-Llama-8B     | 69.7%   | 33.8%  | 52.7%   | 15.4%  | 35.0%   | 4.1%   |
> | QwQ-32B                | 66.0%   | 28.2%  | 48.0%   | 13.3%  | 29.6%   | 5.3%   |
>
>
> In the revised version of our paper, we will include both comprehensive token selection visualizations and the full embedding-based redundancy analysis in the appendix. We believe this will help readers more clearly appreciate the effectiveness of RPC in eliminating redundancy while preserving critical reasoning information.
>
> Once again, we sincerely thank the reviewer for this helpful suggestion.
>
>
> **[Response SCoz-3] Weakness 3: Vague Figure 6**
>
> Thank you for bringing this to our attention. In the revised version of the paper, we will update Figure 6 with a higher-resolution version to enhance readability.
>
>
> **Example Scoz-1 Redundant reasoning path and selected tokens in bold**
>
>
> **[BEGINNING OF EXAMPLE]** **Okay, so I have this problem here about a hyperbola and**  a rhombus inscribed in it. The hyperbola equation is **given as x²/20 - y²/24 = 1. The**  rhombus ABCD has its diagonals intersecting at the origin, and I need to find the greatest real number that is less than BD² for **all**  such rh **ombi. Hmm, okay,**  let me break this down step **by step.**
>
> …
> **Therefore, the problem reduces to points A** (x1,y1) and B(x2,y2) on the **hyperb** ola such that x1x2 + y1y2 =0, and we need to find the minimal BD² (Wait, no, the question says "the greatest real number that is less than BD² for all such rhombi". So actually, we need to find the infimum of BD², but the question is asking for the greatest real number less than all BD². Hmm, maybe the minimal BD²? Or maybe it's the supremum? Wait, no. Wait, "the greatest real number that is less than BD² for all such rhombi". So, in other words, find the infimum of BD², but such that this number is less than all possible BD². Therefore, the infimum is the greatest lower bound, so the answer would be the infimum. But maybe BD² can be **made arbitrarily small? Probably not, because the hyperbola is fixed. Wait,**  but perhaps BD² has a minimum value? Let me think.
>
> Alternatively, maybe BD² can be as small as possible, but maybe there is a lower bound. Wait, but since the hyper **bola is symmetric, maybe BD** ² can be made **as small as approaching some value. Hmm,**  perhaps I **need to parametrize the points.**
>
> …
>   **Let me suppose that OA is (x** ,y) and OB is (u,v). Since OA and OB are perpendicular, **then u = k y, v =**  -k **x for some scalar k. Wait, because if**  (x,y) and (u,v) are perpendicular, then (u,v) is a scalar multiple of (-y, x) or (y, -x). **Let me check:**
>
>   **If OA**  is (x,y), then a perpendicular vector would be (-y, x) or (y, -x). So, (u,v) must be a scalar multiple of that **.**  Let me suppose (u,v) = λ*(-y, x). Then, substituting into the hyperbola equation for B:
>
> u²/20 - v²/24 = ( (λ² y²)/20 ) - ( (λ² x²)/24 ) = λ² ( y²/20 - x²/24 ) **=1.**
>
>   **Wait, but**  the hyperbola equation for point A is x²/20 - y²/ **24 =1, so y²** /24 = (x²/ **20 -1). Hmm, so let me rearrange that.**
>
>   **Wait, let me see:**
>
>   **Given (u** ,v) = λ*(-y, x). Then, substituting into the hyperbola equation:
>
> u²/20 - v²/24 = (λ² y²)/20 - (λ² x²)/24 = λ² ( y²/20 - x²/24 ) **=1.**
>
>   **But from point A's equation, x²/** 20 - y²/2 **4 =1, which can be rearranged to**  y²/24 = (x²/20 -1). Therefore **, y² =24*(** x²/2 **0 -1). Let me plug**  that into the expression above.
>
> So, y²/20 - x²/24 = (24*(x²/20 -1))/20 - x²/24.
>
> Wait, let me compute that **step by step.**
>
>   **First, compute**  y²/20:
>
> y²/20 = [24*(x²/20 -1)] /20 = 24/(20)*(x²/20 -1) = (24/20)*(x²/20 -1  **)**
>
>   **Wait, perhaps this is getting too convoluted. Let me think differently.**
>
>   **Let me denote S = u² + v². Since BD squared is 4** S, **we**  need **to find**  the minimal value **of**  S given that (u,v) lies on the hyperbola and (u,v) is perpendicular to (x,y) which is also on the **hyperbola.**
>
>   **Alternatively, since**  (u,v) is along the direction perpendicular to (x,y), let me parametrise (u,v) as a scalar multiple of (- **y, x).**  So **let me**  write u = **-** λ y, v = λ x **. Then, substit** uting into the hyperbola equation for **B** :
>
> u²/20 - v²/24 = (λ² y²)/20 - (λ² x²)/24 = λ² ( y²/20 - x²/24 ) =1  **.**
>
>   **From point A's hyperbola equation,**  x²/20 - y²/ **24 =1, so rearranged: y²/** 24 = x²/ **20 -1, so y² =24*(** x²/2 **0 -1). Let me substitute that into the expression above.**
>
>   **First, compute**  y²/20 - x²/24:
>
> y²/20 = [24*(x²/20 -1)] /20 = (24/20)*(x²/20 -1) = (6/5)*(x²/20 -1)
>
> x²/24 **is just x²/24.**
>
>   **Therefore, y** ²/20 - x²/24 = (6/5)(x²/20 -1) - x²/24
>
>  **Let me compute this term:** ... **[END OF EXAMPLE]**

---

> > ### Comment · Reviewer_SCoz · 2025-08-05
> >
> > Thank the authors for the detailed responses and additional experiments. They have solved my concerns, and I will raise my score accordingly.

---

> > > ### Author Response · Authors · 2025-08-05
> > >
> > > We appreciate your positive feedback and raising the score.

---

### Note · Authors · 2025-08-12

We sincerely thank the reviewers for their constructive feedback. It has prompted valuable discussions on the proposed RPC.
In this final remark, we would like to highlight three key points from the discussion:

First, there was a discussion on how effective RPC is in reducing redundancy. To address this, we presented visualizations of generated output sequences highlighting the tokens retained after applying RPC. These examples clearly show that RPC effectively eliminates redundancy by preserving only intermediate conclusions essential for reasoning progress and recently generated tokens. Furthermore, for a quantitative perspective, we extended our analysis by applying embedding-based cosine similarity between sentences to compare redundancy in the generated outputs before and after applying RPC. The analysis results show that RPC significantly reduces semantically redundant content in generated outputs.

Second, the difference between evicting generated tokens versus input tokens was discussed. Previous methods such as SnapKV are designed to compress only long input prompts, and thus do not address the KV cache growth from generated outputs. While methods like H2O can evict tokens during generation, they are primarily tuned for scenarios with long inputs and short outputs, and their eviction criteria are not tailored to the evolving importance patterns in reasoning LLMs. In contrast, RPC is specifically designed for long reasoning paths, applying periodic re-evaluation and semantic sparsity exploitation to generated tokens.

Lastly, questions were raised about the practicality of hyperparameter search for the compression interval P and selector window size R. We clarified that their selection is straightforward and does not require exhaustive tuning, as the trade-offs between accuracy and efficiency remain consistent across tasks.

All the feedback from reviewers has helped improve the clarity and completeness of our work. All additional analyses conducted during the discussion will be incorporated into the final version of our manuscript.

We thank the reviewers once again for their careful and thoughtful feedback.

---

### Decision · Program_Chairs · 2025-09-17

**Decision:**

Accept (poster)

**Comment:**

The paper focuses on making LLM reasoning more practical and introduces a KV cache compression approach (RPC) that takes advantage of the redundancy and semantic sparsity of reasoning paths. The proposed method is shown to achieve significant compression with minimal impact on accuracy.

The approach is nicely motivated and intuitive. The various experiments and ablation studies convincingly demonstrate the relevance of the method. In particular the reviewers and AC all appreciate the additional results to address the concerns on output length vs frequency of compression, hyperparameter selection, additional analysis to showcase how RPC effectively preserves valuable content while removing redundant information, additional analysis in support of the the use of attention-based metric.

We strongly urge the authors to follow up on their promise to incorporate these very valuable materials.